# Multivariate Decoding and Drift-Diffusion Modeling Reveal Adaptive Control in Trilingual Comprehension

**DOI:** 10.3390/brainsci15101046

**Published:** 2025-09-26

**Authors:** Yuanbo Wang, Yingfang Meng, Qiuyue Yang, Ruiming Wang

**Affiliations:** 1School of Psychology, Fujian Normal University, Fuzhou 350117, China; 2Philosophy and Social Science Laboratory of Reading and Development in Children and Adolescents, Ministry of Education, and Center for Studies of Psychological Application, School of Psychology, South China Normal University, Guangzhou 510631, China; 3School of Educational Science and Technology, Anshan Normal University, Anshan 114007, China

**Keywords:** trilingualism, dual-language contexts, event-related potentials, drift-diffusion modeling, language comprehension

## Abstract

Background/Objectives: The Adaptive Control Hypothesis posits varying control demands across language contexts in production, but its role in comprehension is underexplored. We investigated if trilinguals, who manage three dual-language contexts (L1–L2, L2–L3, L1–L3), exhibit differential proactive and reactive control demands during comprehension across these contexts. Methods: Thirty-six Uyghur–Chinese–English trilinguals completed an auditory word-picture matching task across three dual-language contexts during EEG recording. We employed behavioral analysis, drift-diffusion modeling, event-related potential (ERP) analysis, and multivariate pattern analysis (MVPA) to examine comprehension efficiency, evidence accumulation, and neural mechanisms. The design crossed context (L1–L2, L2–L3, L1–L3) with trial type (switch vs. repetition) and switching direction (to dominant vs. non-dominant language). Results: Despite comparable behavioral performance, drift-diffusion modeling revealed distinct processing profiles across contexts, with the L1–L2 context showing the lowest comprehension efficiency due to slower evidence accumulation. In the L1–L3 context, comprehension-specific proactive control was indexed by a larger P300 and smaller N400 for L1-to-L3 switches. Notably, no reactive control (switch costs) was observed across any dual-language context. MVPA successfully classified contexts and switching directions, revealing distinct spatiotemporal neural patterns. Conclusions: Trilingual comprehension switching mechanisms differ from production. Reactive control is not essential, while proactive control is context-dependent, emerging only in the high-conflict L1–L3 context. This proactive strategy involves allocating more bottom-up attention to the weaker L3, which, unlike in production, enhances rather than hinders overall efficiency.

## 1. Introduction

With approximately 43% of the world’s population being bilingual and 17% multilingual [1], understanding cognitive control is crucial. Bilinguals and trilinguals automatically activate non-target languages during target language processing [2,3,4], creating cross-language interference that necessitates sophisticated control mechanisms.

Theoretical understanding has predominantly focused on production [5,6], but comprehension control remains poorly understood, with emerging evidence suggesting fundamental differences between modalities [7,8,9]. Auditory word recognition itself engages a distributed neural network, from the auditory ventral stream for phonemic processing [6] and temporal regions for prosodic analysis [10] to even the ventral occipitotemporal cortex for accessing orthographic and semantic information from spoken words [11]. Recent trilingual research confirms that production and comprehension involve distinct control mechanisms [12].

Language control involves two primary mechanisms: reactive control, which resolves conflict after it occurs, and proactive control, which prevents it [5,13]. While models like the Bilingual Interactive Activation (BIA) framework account for bilingual processing [14,15], their application to trilingual comprehension is theoretically and empirically underexplored.

Empirical evidence for comprehension control is complex and inconsistent. Unlike in production, comprehension studies report mixed results for reactive control, with some finding no language switch costs [16] and others finding facilitation effects [17]. Proactive control markers, such as language dominance reversal effects, are also typically absent in comprehension [18,19]. Even with more reliable trilingual measures like n-2 repetition costs [20]—an indicator of reactive inhibitory control—results remain contradictory. The n-2 repetition cost refers to the finding that reaction times are slower in the third trial of an A-B-A sequence (where A, B, and C are different languages) compared to the third trial of a C-B-A sequence. This cost is thought to reflect the lingering inhibition of language A, which was suppressed to allow for the production of language B in the preceding trial. While some studies observe these costs only for the weakest language [21], others find no significant costs at all [12].

The Adaptive Control Hypothesis provides a framework for these complexities, proposing that trilinguals navigate three distinct dual-language contexts (L1–L2, L2–L3, L1–L3), each imposing different control demands [22]. In the present study, the term “context” refers specifically to the language pair in use, rather than broader sociolinguistic or cultural dimensions. This aligns with findings that different language contexts recruit distinct neural mechanisms [23,24]. Furthermore, factors such as the medium of instruction—where L2 may form a cooperative link with L3 [3,25,26]—and orthographic similarity [27] add further complexity to trilingual language processing.

Multilingualism, especially trilingualism, presents unique challenges in cognitive control, yet our understanding of the underlying neural mechanisms in trilinguals remains limited [28]. Research has largely focused on language production, with less attention paid to comprehension, which may involve distinct control mechanisms. A significant gap is the lack of studies on how different dual-language contexts—such as L1–L2, L2–L3, and L1–L3—affect comprehension efficiency. While bilingual research highlights the roles of proficiency and context in production, it is unclear if these patterns hold for comprehension. Specifically, the relationship between proactive (anticipatory) and reactive (conflict-resolution) control in trilingual comprehension is poorly understood.

Our overarching aim is to investigate whether the language control mechanisms in comprehension switching are distinct from those in production switching. To achieve this, we first examine whether a higher demand for conflict monitoring impairs overall processing efficiency in comprehension, as it does in production [29,30]. By comparing the L1–L2 context (lower conflict) with the L1–L3 context (higher conflict), we test the prediction that overall efficiency will be lower in the L1–L3 context if the mechanisms are similar. Second, we investigate if proactive control in comprehension mirrors that in production by testing for a reversal of the language dominance effect. This would be reflected by slower overall processing times for the dominant language, a lower drift rate for “switch to dominant language” trials, and a larger N2 amplitude for such switches. Finally, we assess whether reactive control mechanisms are comparable by looking for asymmetrical language switching costs. We hypothesize that such costs will be asymmetrical in the L1–L3 and L2–L3 contexts, accompanied by longer non-decision times and larger LPC amplitudes for switch trials compared to repetition trials.

This multi-method approach provides a comprehensive view. Multivariate pattern analysis (MVPA) analyzes spatiotemporal patterns across the entire EEG signal, uncovering subtle neural differences and the overlap of control mechanisms that univariate methods might miss. Drift-diffusion modeling (DDM) models behavioral data to estimate parameters like drift rate (evidence accumulation speed) and non-decision time (sensory/motor processes), offering a more detailed understanding of cognitive processes than simple reaction time analysis.

We operationally define proactive control as preparatory processes to prevent conflict, measured by language dominance reversal effects and N2 amplitude. Reactive control is defined as conflict resolution after competition is detected, measured by language switching costs, LPC amplitude (lemma-level inhibition), and N400 amplitude (semantic conflict), consistent with prior work [31,32,33].

Recent research connects DDM parameters to language control [34,35,36,37,38]. Based on this, we propose that proactive control is reflected by decreased drift rate and an increased decision threshold, while reactive control is primarily reflected by increased non-decision time and reduced response bias. We will also conduct exploratory analyses on how event-related potential (ERP) amplitudes (N2, N400, LPC) may influence these DDM parameters.

Previous research on Uyghur–Chinese bilinguals shows that bilingual advantages stem from flexible monitoring strategies rather than superior inhibitory capacity [39], and that cross-linguistic influence is guided by processing efficiency and language-specific structures [40]. However, while these studies provide crucial insights into cognitive control in this population, they did not investigate the mechanisms of language comprehension switching across different dual-language contexts. To address this gap, we test these questions among Uyghur–Chinese–English trilinguals, who show comparable L1–L2 proficiency but lower L3 proficiency, using an auditory word-picture matching task.

## 2. Materials and Methods

### 2.1. Participants

Thirty-six right-handed undergraduates (15 male, 21 female; age range: 18–23 years; *M* = 20.10, *SD* = 1.46) took part for monetary compensation. Inclusion criteria were no acquisition of a fourth language, no immigration background, and normal or corrected-to-normal vision. The exclusion of individuals with any fourth-language acquisition was implemented to maintain a controlled experimental design, ensuring that observed effects could be specifically attributed to the trilingual (L1–L2–L3) system without confounds from more extensive multilingual experience. The study was approved by the South China Normal University IRB (IORG NO.0011738; Approval Code: SCNU-PSY-2022-112; 13 December 2022). All participants provided written informed consent.

#### 2.1.1. Power Analysis

Following Brysbaert [41], we set the minimum meaningful effect for the context main effect at Cohen’s *f* = 0.20 (*α* = 0.05, power = 0.80, assumed within-subject correlation = 0.50) for a design with 12 within-subject conditions. A priori power analysis in G*Power 3.1 [42] indicated a required sample of 24; we recruited 36 to ensure adequate power.

#### 2.1.2. Language Background and Self-Reported Proficiency

Participants completed the Language and Social Background Questionnaire (LSBQ; [43]), rating proficiency in Uyghur (L1), Chinese (L2), and English (L3) across four domains on 10-point scales and usage frequency on 5-point scales (see Table 1).

In this study, “Age of acquisition” refers to the specific age at which participants began systematic exposure to and learning of each language. Age of acquisition differed across languages, *F*(2, 70) = 74, *p* < 0.001, *η*^2^ = 0.766: Uyghur was acquired earliest (*M* = 0.68 years), followed by Chinese (*M* = 5.70) and English (*M* = 11.87). Post hoc tests showed all pairwise differences were significant (all *ps* < 0.001; Cohen’s *ds* > 1.85).

“Exposure” denotes the time spent living in regions where the language is predominantly spoken as a community language. For participants in this study, L3 English exposure = 0 indicates no residency in English-speaking countries, such as the UK or USA, despite having received formal English education in school settings. Languages also differed in exposure duration, *F*(2, 70) = 76, *p* < 0.001, *η*^2^ = 0.570, and in current usage, *F*(2, 70) = 112, *p* < 0.001, *η*^2^ = 0.681, confirming the proficiency/experience hierarchy L1 (Uyghur) > L2 (Chinese) > L3 (English).

“Usage” was calculated by subtracting the age of acquisition from the current age, excluding periods of non-use. Both “Home” and “Social” usage were rated on a 0–4 scale, where 0 indicates no usage and 4 indicates exclusive usage. The LSBQ separates Home Use (family/domestic) from Social Use (community). Home Use differed by language, *F*(2, 70) = 53.60, *p* < 0.001, *η*^2^ = 0.505, with Uyghur highest, then Chinese, then English. For Social Use, *F*(2, 70) = 98.10, *p* < 0.001, *η*^2^ = 0.651, Chinese exceeded Uyghur (*p* < 0.001, Cohen’s *d* = 3.062), while English did not differ significantly from Uyghur. This pattern reflects Chinese as the national common language and Uyghur as the heritage language.

“Learning contexts” define acquisition environments (home, school, both). Participants learned their heritage L1 (Uyghur) at home and the national L2 (Chinese) at school. “Medium-of-instruction” refers to the language used for teaching in school. In this study, L3 (English), a foreign language, was learned with L2 (Chinese) serving as the medium of instruction.

#### 2.1.3. Objective Proficiency Verification

Because self-ratings can vary across cultures [44], we administered the Multilingual Naming Test (MINT; [45]), a standardized 68-item picture-naming measure. The MINT is a standardized, cross-linguistic tool used to assess lexical retrieval abilities in multilingual individuals. The 68-item version consists of culturally neutral, high-frequency images, minimizing linguistic and cultural biases. Participants name each item as quickly and accurately as possible, and scoring is based on the number of correct responses, with a maximum score of 68. Responses are only accepted if they are the conventional names for the depicted objects in the target language, and partial credit is not given. This test provides a quantitative measure of expressive vocabulary and naming performance in multilingual individuals. MINT scores differed by language, *F*(2, 70) = 287, *p* < 0.001, *η*^2^ = 0.845: Uyghur and Chinese did not differ (*p* = 0.942, *d* = 0.077), whereas English scores were significantly lower than both (all *ps* < 0.001; *ds* > 4.85), corroborating the pattern L1 ≈ L2 > L3.

### 2.2. Materials and Procedure

Forty-eight pictures were drawn from the International Picture Naming Project [46] with corresponding words in Uyghur (L1), Chinese (L2), and English (L3). Morphologically complex and compound words were excluded. Morphologically complex words, composed of multiple morphemes (e.g., prefixes, suffixes), were excluded to ensure experimental control. This step avoided confounding variables from cross-linguistic differences in morphological processing. It allowed the analysis to focus specifically on language control mechanisms by removing the additional cognitive demands associated with complex word decomposition and formation. Twenty-five trilingual validators (non-participants) rated vocabulary familiarity and picture–word matching on 7-point scales. Ratings were high and did not differ by language for familiarity, *F*(2, 428) = 1.73, *p* = 0.181; matching accuracy, *F*(2, 428) = 0.39, *p* = 0.677; or syllable count, *F*(2, 428) = 3.04, *p* = 0.062.

The experiment comprised three dual-language contexts (L1–L2, L2–L3, L1–L3), each using the same auditory word–picture matching task. Crossing context (3) with trial type (4)—switch to non-dominant, switch to dominant, non-dominant repetition, dominant repetition—yielded 12 within-subject conditions, with 30 trials per condition.

The design distinguished between switch (different languages) and repetition (same language) trials. Language dominance was defined by age of acquisition: L1 (Uyghur) in L1–L2/L3 contexts, and L2 (Chinese) in the L2–L3 context. Switches were to-dominant or to-non-dominant. Switching costs were Switch RT—Repetition RT. Language dominance effects were the RT difference between the overall mean RTs for non-dominant and dominant languages (including both switch and repetition trials).

Auditory stimuli were recorded by three trilingual speakers and assigned according to a Latin square to control for speaker effects. Recordings were amplitude-normalized to 70 dB in Praat (mean duration = 574 ± 164 ms).

#### Experimental Procedure

The task was presented in three counterbalanced blocks corresponding to the three contexts, using E-Prime 3.0. Each trial began with a fixation cross (500 ms), followed by an auditory word on a blank screen. After word offset, the blank screen continued for 600–900 ms, then a picture appeared for 2000 ms. Participants made match/mismatch judgments; response-key mapping was counterbalanced across participants (see Figure 1).

### 2.3. Data Acquisition and Preprocessing

#### 2.3.1. EEG Recording

EEG was recorded from 32 Ag/AgCl electrodes (Quik-Cap, Compumedics, Melbourne, Australia) using SynAmps2 amplifiers and Curry 8.0.6 software. Signals were sampled at 1 kHz, online-referenced to the left mastoid, and band-pass filtered at 0.05–100 Hz. Electrode impedances were kept below 5 kΩ.

#### 2.3.2. Preprocessing

Analyses were conducted in MATLAB R2020a with EEGLAB [47]. Incorrect responses and filler trials (5.75%) were removed; filler trials were the first trial of each block and were not counted as switch or repetition. Signals were low-pass filtered at 30 Hz and re-referenced to the bilateral mastoids. Continuous data were epoched from −200 to 1000 ms relative to the onset of the second auditory word in the trial, with baseline correction applied from −200 to 0 ms. Epochs exceeding ±80 μV were rejected, and ocular artifacts were removed using Independent Component Analysis.

### 2.4. Data Analyses

#### 2.4.1. Analytical Framework

Our analytical approach employed four complementary methods for a comprehensive analysis across behavioral, computational, and neural levels. We first analyzed behavioral data (reaction times, accuracy), then used drift-diffusion modeling (DDM) to decompose decision processes into cognitive components like evidence accumulation and non-decision time, revealing hidden processing differences. At the neural level, event-related potentials (ERPs) provided precise temporal information, complemented by multivariate pattern analysis (MVPA) to detect distributed neural patterns distinguishing between conditions.

Finally, multiple regression linked ERP components to DDM parameters to clarify their mechanistic relationship. This convergent strategy addresses inconsistencies in prior research [12,21] by examining data through multiple lenses, allowing us to distinguish genuine null effects from compensatory processes that mask underlying processing differences.

#### 2.4.2. Behavioral Analysis

Data preprocessing involved excluding incorrect responses and trials with reaction times exceeding ±2.5 standard deviations from each participant’s mean. Switch costs were calculated as the reaction time difference between switch and repetition trials (RTswitch—RTrepetition).

Statistical analyses employed linear mixed-effects models using the lme4 package in R [48]. The experimental design consisted of a 2 (trial type: switch, repetition) × 2 (switching direction: to dominant language, to non-dominant language) × 3 (context: L1–L2, L2–L3, L1–L3) within-subjects factorial design. Separate models were fitted for accuracy and reaction time data to account for their different distributional properties.

For accuracy data, a generalized linear mixed-effects model with binomial distribution was employed:glmer(Accuracy ~ trial_type × switching_direction × context + (1|Participant) + (1|Item), family = binomial)

For reaction time data, a linear mixed-effects model was used:lmer(Reaction_Times ~ trial_type × switching_direction × context + (1|Participant) + (1|Item))

Both models included fixed effects for trial type, switching direction, context, and all their interactions. Random intercepts for participants and items were included to account for individual differences and stimulus-specific variance, respectively.

#### 2.4.3. Event-Related Potential (ERP) Analysis

We analyzed event-related potentials (ERPs) from −200 to +1000 ms relative to auditory word onset. Statistical models tested the main effects of trial type, switch direction, and context, along with all two- and three-way interactions, using the same 2 × 2 × 3 within-subjects design as in the behavioral analyses.

Analyses focused on two regions of interest based on prior language-switching work: frontal electrodes (FCz, Fz) and central–parietal electrodes (Cz, CPz, Pz). We examined five components commonly implicated in language processing and switching: (i) P200 (150–200 ms), an early auditory response linked to attention and perceptual processing at the phonological [49,50]—rather than lexical [51] —level; (ii) N200 (200–300 ms), associated with conflict monitoring and often linked to anterior cingulate and medial prefrontal activity [13,52,53]; (iii) P300 (300–400 ms), indexing allocation of attentional resources, typically maximal over parietal and temporoparietal regions [54]; (iv) N400 (400–500 ms), sensitive to semantic processing and commonly observed over temporal sites [55,56,57,58], particularly the left middle temporal gyrus; and (v) the late positive complex (LPC; 500–800 ms), reflecting late reanalysis and control processes—specifically, reconfiguring stimulus–response mappings to regain access to previously inhibited lexical representations—with generators often reported in parietal and medial prefrontal regions [59,60].

For each condition, we evaluated raw waveforms and condition-difference waves against baseline using two-tailed *t*-tests at each time point. To control for multiple comparisons, we applied cluster-based permutation testing with 1000 iterations.

#### 2.4.4. Multivariate Pattern Analysis (MVPA)

MVPA used the Amsterdam Decoding and Modeling toolbox (ADAM; [61]) with linear discriminant analysis to decode condition information from −200 to 1000 ms. Performance was estimated using the area under the receiver operating characteristic curve (AUC) with 10-fold cross-validation run separately for main effects of the context, trial type, and switching direction. Significance was assessed via cluster-based permutation tests (*p* < 0.05). To interpret classifier weights, activation patterns were derived using the Haufe transform [62] and z-scored across electrodes per participant to compare spatial distributions across conditions. Temporal generalization [63] trained classifiers at each time point and tested them at all other time points to assess the stability of neural codes; cross-validation folds were kept independent of temporal generalization.

#### 2.4.5. Drift-Diffusion Model (DDM) Analysis

Drift-diffusion models were fitted using the Dynamic Models of Choice (DMC) package [64] in R, estimating five parameters: drift rate (for match [*d.s1*] and mismatch [*d.s2*] responses), boundary separation (a), starting point (z), and non-decision time (Ter). This setup allowed us to isolate language switching effects from response switching effects. The model included the factors of context, switching direction, and trial type.

Model fitting and validation followed a hierarchical Bayesian approach [65,66]. We confirmed the model’s reliability through a series of rigorous diagnostic checks:

Convergence: All model chains successfully converged, as indicated by Gelman-Rubin statistics remaining well below the 1.10 criterion [67] (max R-hat = 1.04). Visual inspection of the trace plots further confirmed stable chain mixing with no discernible trends.

Goodness-of-Fit: Posterior predictive checks showed an excellent fit, confirming that the model could accurately reproduce the key features of the observed reaction time distributions (e.g., their central tendency, variance, and skewness) across all experimental conditions.

Identifiability: Parameter-recovery simulations demonstrated that all model parameters could be accurately retrieved from simulated data (all recovery correlations *r* > 0.90, *ps* < 0.001), ensuring the model was identifiable.

These comprehensive checks provide strong evidence for the reliability and validity of our DDM parameter estimates.

#### 2.4.6. Multiple Regression Linking ERPs and DDM

To test whether ERP amplitudes predict DDM parameters, five linear regressions (using MATLAB’s fitlm function) were run—one per DDM parameter—as outcomes, with LPC, N200, N400 amplitudes and context as predictors. Interactions (Context × N200; Context × N400) tested context-specific relations. Assumptions were checked via residual diagnostics: Pearson residuals vs. fits/predictors (homoscedasticity, linearity), normal probability plots (normality), Durbin–Watson tests (~2.0; independence), and variance inflation factors (all VIF < 5). Influential points were screened using leverage > 3 × (k + 1)/n and Cook’s *D* > 1; no cases exceeded thresholds. *p*-values were adjusted with the Bonferroni–Holm procedure. Adjusted response and interaction plots visualized effects while holding other predictors at their fitted values.

## 3. Results

### 3.1. Results of Reaction Time Data

As shown in Table 2 and Figure 2, mixed-effects model analysis revealed the following patterns. For switching direction, a significant main effect emerged, *t* = 3.724, *p* < 0.001, Cohen’s *d* = 0.309, with faster reaction times when switching to dominant languages (*M* = 1558 ms) compared to non-dominant languages (*M* = 1573 ms). For context, no significant main effect was observed, with comparable reaction times across L1–L2 (*M* = 1563 ms), L1–L3 (*M* = 1569 ms), and L2–L3 (*M* = 1565 ms) contexts (all pairwise comparisons *p* > 0.149). While these differences may appear small, they align with recent findings in the field [12]. Language control effects in comprehension are more subtle than in production. Therefore, sophisticated analytical approaches are required to detect these effects.

For trial type, no significant main effect emerged between switch (*M* = 1563 ms) and repetition (*M* = 1569 ms) trials, *t* = 1.89, *p* = 0.059, Cohen’s *d* = 0.051. Regarding the switching direction × context interaction, it was not significant overall, though context-specific patterns emerged. In the L2–L3 context, switching to non-dominant languages produced significantly slower responses (*M* = 1575 ms) than switching to dominant languages (*M* = 1555 ms), *t* = 3.027, *p* = 0.042, Cohen’s *d* = 0.337. No significant differences were observed in L1–L2 (*p* = 0.378) or L1–L3 (*p* = 0.097) contexts.

For the switching direction × trial type interaction, no significant interaction emerged. Switch costs were comparable between dominant language conditions (switch: *M* = 1562 ms; repetition: *M* = 1554 ms, *p* = 0.065) and non-dominant language conditions (switch: *M* = 1575 ms; repetition: *M* = 1571 ms, *p* = 0.396).

For the context × trial type interaction, no significant interaction was found, with comparable switch costs across L1–L2 (3 ms, *p* = 0.533), L1–L3 (1 ms, *p* = 0.874), and L2–L3 (14 ms, *p* = 0.182) contexts.

Finally, the switching direction × context × trial type interaction was not significant. Switch costs for dominant versus non-dominant language conditions showed no significant differences within any context: L1–L2 (2 ms vs. 5 ms, *p* = 0.730), L2–L3 (19 ms vs. 8 ms, *p* = 0.300), and L1–L3 (4 ms vs. −2 ms, *p* = 0.633).

Critically, the absence of large behavioral differences does not indicate absence of control mechanisms. Instead, it suggests that trilingual comprehension relies on efficient, automatized control processes that operate below the threshold of gross behavioral detection but can be revealed through computational decomposition of the decision process.

### 3.2. Results of Accuracy Data

As shown in Table 3 and Figure 3, logistic mixed-effects model analysis revealed no significant main effects or interactions for accuracy data. For trial type, no significant main effect emerged between switch (*M* = 0.974) and repetition (*M* = 0.974) trials, *z* = −0.014, *p* = 0.989, Cohen’s *d* = 0.011. For switching direction, no significant main effect was observed between switching to dominant (*M* = 0.976) versus non-dominant (*M* = 0.971) languages, *z* = 0.979, *p* = 0.327, Cohen’s *d* = 0.035. For context, no significant main effect was found across contexts, with comparable accuracy in L1–L2 (*M* = 0.970), L1–L3 (*M* = 0.976), and L2–L3 (*M* = 0.974) contexts (all pairwise comparisons *p* > 0.391). The trial type × switching direction interaction was not significant, with accuracy being comparable between switching directions in both switch trials (dominant: *M* = 0.976; non-dominant: *M* = 0.971, *p* = 0.808) and repetition trials (dominant: *M* = 0.976; non-dominant: *M* = 0.972, *p* = 0.875). Similarly, the trial type × switching direction × context interaction was not significant. Switch costs (calculated as accuracy differences) for dominant versus non-dominant language conditions showed no significant differences within any context: L1–L2 (−0.45% vs. −0.13%, *p* = 0.407), L2–L3 (0.003% vs. 0.19%, *p* = 0.648), and L1–L3 (0.50% vs. −0.10%, *p* = 0.153). Accuracy performance remained consistently high across all experimental conditions (range: 0.959–0.983).

### 3.3. Drift Diffusion Model Results

To examine the underlying cognitive processes in trilingual language switching, we analyzed five key diffusion model parameters using linear mixed-effects models with context type, switching direction, and trial type as fixed effects and participant intercepts as random effects.

#### 3.3.1. Drift Rate Analysis for Match Responses (*d.s1*)

The drift rate analysis examined the speed of evidence accumulation for stimuli requiring match responses, providing a direct measure of comprehension efficiency uncontaminated by speed-accuracy trade-offs or response preparation processes.

Context effects on evidence accumulation showed a discernible pattern. As depicted in Figure 4, the mixed linear model indicated a statistically significant effect on *d.s1*. Processing speed was slower in the L1–L2 dual-language context (*M* = 3.17) than in the L1–L3 dual-language context (*M* = 4.21, *t* = −2.48, *p* = 0.043, Cohen’s *d* = −0.541). The observed reduction in drift rate was not apparent in the behavioral data, as it was offset by compensatory adjustments in other decision parameters. In the L1–L2 context, despite both languages being highly proficient, the simultaneous high-level activation of two dominant language systems creates a “processing bottleneck” where evidence accumulation becomes sluggish due to intense cross-linguistic competition. This does not support the prediction from the Bilingual Interactive Activation model that balanced bilingual activation decreases processing demands. Instead, our findings suggest that balanced activation paradoxically increases processing costs. One possible explanation for this counterintuitive finding is that the L1–L2 context may lack proactive control mechanisms, thereby reducing language comprehension efficiency. When two languages are equally dominant and frequently co-activated, the cognitive system may fail to implement the selective attention and inhibitory control processes that are automatically engaged in more asymmetric language contexts [68]. This absence of proactive control allows cross-linguistic interference to persist throughout the comprehension process [69,70], creating the observed processing bottleneck despite high proficiency in both languages.

However, no significant differences were observed between L1–L2 and L2–L3 contexts (*M* = 3.62, *t* = −1.09, *p* = 0.524, Cohen’s *d* = −0.238), nor between L2–L3 and L1–L3 contexts (*t* = 1.39, *p* = 0.355, Cohen’s *d* = 0.303). These findings indicate that participants exhibited the fastest processing speed for word-picture matching judgments in the L1–L3 dual-language context.

With respect to trial type, as shown in Figure 5, the mixed linear model revealed no significant main effect on *d.s1* Switch trials (*M* = 3.55) did not differ significantly from repeat trials (*M* = 4.21, *t* = 1.42, *p* = 0.168, Cohen’s *d* = 0.308).

Concerning switch direction, Figure 6 demonstrates that the mixed linear model showed no significant main effect on *d.s1* No significant difference was found between switching to the dominant language (*M* = 4.26) and switching to the non-dominant language (*M* = 4.13, *t* = −0.388, *p* = 0.701, Cohen’s *d* = −0.073).

#### 3.3.2. Drift Rate Analysis for Mismatch Responses (*d.s2*)

The drift rate analysis for mismatch responses examined the speed of evidence accumulation for stimuli requiring rejection responses (i.e., when participants pressed the “no” key indicating word-picture mismatch).

Regarding the main effect of language context, the mixed linear model indicated a statistically significant main effect on *d.s2*. Participants’ drift rates for rejection responses were higher in the L1–L2 context (*M* = −3.37) compared to both the L1–L3 context (*M* = −4.22, *t* = 2.25, *p* = 0.040, Cohen’s *d* = 0.612) and the L2–L3 context (*M* = −4.08, *t* = 2.21, *p* = 0.037, Cohen’s *d* = 0.540). This pattern indicates an asymmetry in trilingual processing. When both languages in a context are highly proficient (L1–L2), participants struggle not only to accumulate positive evidence for matches but also to efficiently reject mismatches. This “double deficit” suggests that balanced high-proficiency contexts create processing interference that affects both target detection and distractor rejection processes. This finding suggests that the cognitive advantage typically attributed to balanced bilingualism may come with hidden costs in specific dual-language contexts, where the absence of a clear dominance hierarchy prevents the system from efficiently biasing attention toward the target language.

No significant difference was observed between L2–L3 and L1–L3 contexts (*t* = −0.234, *p* = 0.970, Cohen’s *d* = −0.050).

With respect to trial type, The mixed linear model showed no significant main effect of trial type on *d.s2*. Switch trials (*M* = −3.76) did not differ significantly from repeat trials (*M* = −4.55, *t* = −1.76, *p* = 0.090, Cohen’s *d* = −0.373).

Concerning switch direction, No significant main effect of switch direction was found for *d.s2* Switching to the non-dominant language (*M* = −4.52) did not differ significantly from switching to the dominant language (*M* = −4.75, *t* = 0.648, *p* = 0.522, Cohen’s *d* = 0.128).

#### 3.3.3. Starting Point Analysis (Response Bias)

The starting point (*z* value) represents the bias toward one of the two response alternatives in the decision process.

Regarding the main effect of language context, the mixed linear model indicated a statistically significant effect on the *z* value. Participants exhibited a greater bias toward ‘yes’ responses in the L1–L2 context (*M* = 0.857) compared to the L1–L3 context (*M* = 0.764, *t* = 3.04, *p* = 0.011, Cohen’s *d* = 0.756). No significant differences were observed between L1–L2 and L2–L3 contexts (*M* = 0.804, *t* = 1.49, *p* = 0.425, Cohen’s *d* = 0.411) or between L2–L3 and L1–L3 contexts (*t* = 1.49, *p* = 0.421, Cohen’s *d* = 0.345). These results indicate that participants were more inclined to make “yes” responses in the L1–L2 dual-language context.

With respect to trial type, no significant main effect was found for the *z* value. Switch trials (*M* = 0.035) did not differ significantly from repeat trials (*M* = 0.037, *t* = 0.648, *p* = 0.523, Cohen’s *d* = 0.130).

Concerning switch direction, the mixed linear model showed no significant main effect on the z value. No difference was observed between switching to the non-dominant language (*M* = 0.036) and switching to the dominant language (*M* = 0.037, *t* = −0.412, *p* = 0.683, Cohen’s *d* = 0.092).

#### 3.3.4. Boundary Separation Analysis (Decision Threshold)

Boundary separation (*a* value) represents the amount of evidence required by participants to accept one of the two alternatives in the decision process.

Regarding the main effect of language context, the mixed linear model indicated a statistically significant effect on boundary separation. Participants’ decision thresholds were lower in the L1–L2 context (*M* = 0.666) compared to the L2–L3 context (*M* = 1.318, *t* = −2.194, *p* = 0.032, Cohen’s *d* = −0.647). No significant differences were found between L1–L2 and L1–L3 contexts (*M* = 1.733, *t* = −1.804, *p* = 0.179, Cohen’s *d* = −0.496) or between L2–L3 and L1–L3 contexts (*t* = 0.700, *p* = 0.487, Cohen’s *d* = 0.199).

With respect to trial type, no significant main effect was observed for the *a* value. Switch trials (*M* = 0.59) did not differ significantly from repeat trials (*M* = 1.37, *t* = 1.22, *p* = 0.232, Cohen’s *d* = 0.268).

Concerning switch direction, the mixed linear model showed no significant main effect on the a value. No difference was found between switching to the non-dominant language (*M* = 1.32) and switching to the dominant language (*M* = 1.39, *t* = −0.199, *p* = 0.843, Cohen’s *d* = 0.046).

#### 3.3.5. Non-Decision Time Analysis (Motor and Encoding Processes)

Non-decision time (*Ter*) represents the duration of neurological processes involved in encoding sensory stimuli and executing motor responses.

Regarding the main effect of language context, the mixed linear model indicated a statistically significant effect on *Ter*. Participants’ non-decision times were longer in the L1–L2 context (*M* = 0.361) compared to the L2–L3 context (*M* = 0.257, *t* = 3.12, *p* = 0.009, Cohen’s *d* = 0.677). No significant differences were observed between L1–L2 and L1–L3 contexts (*M* = 0.301, *t* = 1.79, *p* = 0.238, Cohen’s *d* = 0.418) or between L2–L3 and L1–L3 contexts (*t* = 1.30, *p* = 0.603, Cohen’s *d* = 0.259).

With respect to trial type, no significant main effect was found for *Ter*. Switch trials (*M* = 0.489) did not differ significantly from repeat trials (*M* = 0.407, *t* = −1.07, *p* = 0.295, Cohen’s *d* = 0.205).

Concerning switch direction, the mixed linear model showed no significant main effect on *Ter*. No difference was observed between switching to the non-dominant language (*M* = 0.502) and switching to the dominant language (*M* = 0.483, *t* = 0.323, *p* = 0.750, Cohen’s *d* = 0.060).

### 3.4. ERP Results

As shown in Figure 7, cluster-based permutation testing revealed significant main effects for context and switching direction, but not for trial type.

Context effects revealed the neural signature of differential control demands. L1–L2 compared to L2–L3 contexts showed significant differences from 344 to 552 ms after auditory word onset (*p* = 0.002), characterized by reduced P300 and enhanced N400 amplitudes in L1–L2 contexts. This pattern indicates that the L1–L2 context, despite being most practiced, paradoxically requires greater semantic conflict resolution (larger N400). Simultaneously, it allocates fewer attentional resources (smaller P300). The counterintuitive nature of this finding requires careful interpretation. The enlarged N400 in L1–L2 context does not reflect processing difficulty per se, but rather the intensity of cross-linguistic semantic competition when two highly activated language systems compete for lexical access. The reduced P300 suggests that this competition occurs automatically, requiring less conscious attentional control but creating more semantic-level interference.

L1–L2 compared to L1–L3 contexts revealed two significant time windows: 360–562 ms (*p* = 0.005) and 794–946 ms (*p* = 0.017) after auditory word onset. L1–L2 contexts elicited smaller P300, larger N400, and reduced late positive component (LPC) amplitudes relative to L1–L3 contexts. No significant differences were observed between L2–L3 and L1–L3 contexts.

In terms of switching direction, switch to non-dominant language conditions differed significantly from switch to dominant language conditions during 368–492 ms after auditory word onset (*p* = 0.024), with dominant language switches eliciting larger N400 amplitudes.

As for trial type, no significant differences were found between switch and repeat trials across any time windows tested.

As shown in Figure 8. The critical test of proactive control mechanisms emerged in the switching direction analysis. A significant interaction between language context and switching direction revealed that proactive control operates selectively across trilingual contexts.

Regarding the L1–L2 and L2–L3 contexts, no significant differences were observed between switch to dominant language and switch to non-dominant language conditions across any time windows tested.

In contrast, within the L1–L3 context, a striking language dominance reversal occurred (298–504 ms, *p* = 0.006), where switches to the dominant language (L1) elicited enhanced P300 and N400 amplitudes compared to switches to the non-dominant language (L3). This reversal provides direct neural evidence for proactive inhibition of the dominant language to facilitate processing of the weaker language. The absence of this pattern in L1–L2 and L2–L3 contexts reveals the boundary conditions of proactive control in trilingual comprehension. In L1–L2 contexts, comparable proficiency levels eliminate the need for asymmetric inhibition. In L2–L3 contexts, the facilitative medium-of-instruction relationship between L2 and L3 creates cooperative rather than competitive dynamics.

As shown in Figure 9, the three-way interaction between language context, switching direction, and trial type was not significant.

The absence of significant switch costs across all contexts (Figure 9) provides compelling evidence against reactive control mechanisms in trilingual auditory comprehension. This null finding is theoretically significant because it demonstrates that comprehension control operates primarily through proactive preparation rather than reactive conflict resolution.

### 3.5. Multivariate Pattern Analysis Results

#### 3.5.1. Diagonal Decoding Results

One core objective of this study was to determine whether neural signals associated with language comprehension under different dual-language contexts could be uncovered using multivariate pattern analysis. We trained linear discriminant classifiers to distinguish between different experimental conditions based on scalp-wide EEG responses (see Section 2 for details). The classification results revealed significant main effects for context manipulations, with the classifier successfully discriminating between L1–L2 and L2–L3 dual-language contexts (Figure 10A; two-tailed cluster *p* < 0.001 after 1000 iterations).

Classification accuracy increased rapidly from approximately 273 ms post-stimulus onset, remaining above chance for a 27 ms time window, with temporal dynamics closely matching the N2 component observed in the difference wave between L1–L2 and L2–L3 context-evoked ERPs. Similarly, significant classification was achieved between L2–L3 and L1–L3 contexts (Figure 10A; two-tailed cluster *p* < 0.001 after 1000 iterations), with above-chance performance emerging at 253 ms post-stimulus and sustaining for 46 ms, aligning with the N2 component in the corresponding ERP difference wave. The most robust discrimination was observed for L1–L2 versus L1–L3 contexts, which showed reliable classification across multiple time windows (Figure 10A; two-tailed cluster *p* < 0.001 after 1000 iterations), including the P200 component (187–226 ms, 39 ms duration), P300 component (325–422 ms, 97 ms duration), N400 component (454–585 ms, 131 ms duration), and LPCs (638–701 ms, 63 ms duration; 822–868 ms, 46 ms duration), with each period corresponding to the respective ERP components in the L1–L2 vs. L1–L3 difference wave. Regarding switching direction, the classifier successfully decoded switches to dominant versus non-dominant languages (Figure 10A; two-tailed cluster *p* < 0.001 after 1000 iterations), with above-chance classification occurring during an early window (225–271 ms, 46 ms duration) corresponding to the N2 component, and late windows (830–869 ms, 39 ms duration; 936–971 ms, 35 ms duration) corresponding to LPCs in the switching direction difference wave. In contrast, no significant classification was achieved for trial type (switch vs. repetition trials), as the classifier failed to discriminate between switch and repetition conditions throughout the analyzed time period, with classification accuracy falling significantly below chance level during brief periods (39–53 ms and 309–319 ms; two-tailed cluster *p* < 0.001 after 1000 iterations), suggesting systematic misclassification rather than random performance. These results demonstrate that context and switching direction information was clearly encoded in the EEG signals and detectable through multivariate pattern analysis, revealing neural distinctions that remained hidden to classical ERP methodology.

#### 3.5.2. Weight Projection Analysis Results

To identify the neural sources underlying successful MVPA classifications, we computed weight projections by multiplying the obtained weight matrices with the covariance matrices (see Section 2), with the resulting topographical maps in Figure 10B representing the spatial distribution of neural signals contributing to classification accuracy across different experimental conditions.

Significant clusters emerged during 236–336 ms (*p* < 0.001, two-tailed cluster-based permutation test, 1000 iterations) when comparing L1–L2 versus L2–L3 contexts, where L2–L3 contexts showed increased activity in right frontal and right fronto-temporal regions (F4/FC4/FT8) but decreased activity in left temporal and left temporo-parietal junction areas (T7/TP7/CP3/P7; Figure 10B).

Similarly, significant topographical differences were observed during 238–338 ms (*p* < 0.001, two-tailed cluster-based permutation test, 1000 iterations) when comparing L2–L3 versus L1–L3 contexts, with L1–L3 contexts eliciting increased activity in left temporal, left inferior parietal, and left centro-parietal junction regions (T7/CP3/P7), alongside decreased activity in right temporal, right temporo-parietal junction, and right inferior parietal areas (T8/TP8/P8; Figure 10B), while a trend toward increased activity in left inferior parietal regions (P7) was observed for L1–L3 versus L1–L2 contexts, though this did not reach statistical significance (Figure 10B).

Weight projection analysis further revealed significant main effects of switching direction during 208–308 ms (*p* < 0.001, two-tailed cluster-based permutation test, 1000 iterations), where switches to dominant languages, compared to switches to non-dominant languages, elicited increased activity in left parietal and left occipital regions (P3/O1/Oz) but decreased activity in right frontal, right fronto-central, and right fronto-temporal areas (F4/FC4/FT8; Figure 10B). Additionally, significant main effects of trial type emerged during 262–362 ms (*p* < 0.001, two-tailed cluster-based permutation test, 1000 iterations), with repetition trials, compared to switch trials, showing decreased activity in bilateral frontopolar, left frontal, and fronto-central midline regions (FP1/FPz/FP2/F3/Fz/FCz; Figure 10B).

#### 3.5.3. Temporal Generalization Using Classification Across Time Result

To investigate the temporal dynamics of context representations, we employed multivariate pattern analysis (MVPA) with temporal generalization procedures [63] to determine whether dual-language context information persists throughout the trial or dissipates immediately after auditory word presentation. The analysis revealed distinct temporal patterns across different dual-language contexts, with L1–L2 versus L2–L3 classification showing only marginal above-chance accuracy from 250 to 321 ms post-stimulus (*p* > 0.05), indicating brief and unstable context representation during lexical access (Figure 10C). In contrast, L2–L3 versus L1–L3 contexts demonstrated robust classification performance from 227 to 387 ms post-stimulus (*p* < 0.005, two-tailed cluster-based permutation test, 1000 iterations), followed by a secondary peak from 410 to 643 ms (*p* < 0.005, two-tailed cluster-based permutation test, 1000 iterations), suggesting initial sustained context representation with subsequent reactivation during semantic processing stages. Similarly, L1–L2 versus L1–L3 contexts exhibited prolonged above-chance classification from 174 to 582 ms post-stimulus (*p* < 0.005, two-tailed cluster-based permutation test, 1000 iterations), with an additional late reactivation period from 622 to 713 ms (*p* < 0.005, two-tailed cluster-based permutation test, 1000 iterations), indicating extended context maintenance and semantic-stage retrieval. Complementing these context effects, analysis of switching direction revealed a biphasic temporal pattern, with classification of switching to dominant versus non-dominant languages exceeding chance from 421 to 498 ms post-stimulus (*p* < 0.005, two-tailed cluster-based permutation test, 1000 iterations) and again from 843 to 994 ms (*p* < 0.005, two-tailed cluster-based permutation test, 1000 iterations), suggesting initial switching direction encoding followed by semantic-stage reactivation. Notably, temporal generalization analysis failed to reliably decode trial type information (switch versus repetition) throughout the experimental timeline (*p* > 0.05), indicating that trial type representations are transient and do not persist during lexical access, reflecting brief, state-dependent processing rather than sustained representation. These findings collectively demonstrate that context and switching direction information exhibit sustained neural representations with distinct temporal profiles, while trial type effects reflect more ephemeral processing mechanisms, supporting the hypothesis that different aspects of multilingual language control operate through temporally distinct neural mechanisms.

#### 3.5.4. Multiple Regression Analysis of ERP Data and Drift Diffusion Model Parameters

We examined relationships between mean ERP amplitudes (LPC, N200, N400) and diffusion model parameters using multiple linear regression analyses. Given non-significant main effects of switching direction and trial type on diffusion model parameters, analyses focused on language context effects. All analyses controlled for participant-level random effects, with multiple comparisons corrected using the Bonferroni-Holm procedure. Estimated regression parameters are provided in Table 4.

N200 Amplitudes and Drift Rate for “No” Responses

N200 amplitudes showed context-dependent associations with drift rates for negative responses (Figure 11A). Note that N200 is a negative-going component; larger numerical values indicate smaller negative amplitudes and reduced inhibitory control. Drift rates for “no” responses are negative values, with more negative values indicating faster accumulation toward the “no” boundary.

In the L2–L3 context, participants with greater N200 numerical values exhibited significantly higher (less negative) drift rates for “no” responses (*b* = 0.144, *t* = 2.21, *p* = 0.030, *R*^2^ = 0.15), indicating that reduced inhibitory control was associated with slower “no” responses. Conversely, in the L1–L3 context, a marginally significant opposite pattern emerged (*b* = −0.119, *t* = −1.86, *p* = 0.068, *R*^2^ = 0.04), where enhanced inhibitory control was associated with faster “no” response rates. No significant relationship was observed in the L1–L2 context (*b* = 0.023, *t* = 0.89, *p* = 0.376, *R*^2^ = 0.01).

These differential patterns explain the observed context effects on drift rates for negative responses, with the L2–L3 context demonstrating faster “no” response patterns compared to the L1–L2 context (*t* = −2.21, *p* = 0.037, Cohen’s *d* = −0.540) due to enhanced inhibitory demands.

N400 Amplitudes and Decision Boundary

N400 amplitudes demonstrated a significant positive relationship with decision boundary settings, exclusively in the L2–L3 context (*b* = −0.29, *t* = −3.45, *p* < 0.001, *R*^2^ = 0.15; Figure 11B). Note that N400 is a negative-going component; larger numerical values indicate smaller negative amplitudes. Neither the L1–L2 context (*b* = 0.035, *t* = 0.63, *p* = 0.531, *R*^2^ < 0.01) nor the L1–L3 context (*b* = −0.030, *t* = −0.38, *p* = 0.706, *R*^2^ < 0.01) showed significant N400-decision boundary relationships.

Context analysis indicated that the L2–L3 context was associated with larger decision boundaries compared to the L1–L2 context (*t* = −3.04, *p* = 0.011, Cohen’s *d* = −0.756), an effect mediated by increased semantic conflict monitoring in the L2–L3 context condition (i.e., enhanced N400 amplitudes).

LPC Amplitudes and Response Bias

Multiple regression analysis revealed a significant negative relationship between LPC amplitudes and response bias, with context-dependent effects (Figure 11C). In the L1–L3 context, participants with greater LPC amplitudes demonstrated significantly lower response bias (*b* = −0.013, *t* = −2.35, *p* = 0.022, *R*^2^ = 0.18). A similar trend emerged in the L1–L2 context at marginal significance (*b* = −0.011, *t* = −1.93, *p* = 0.057, *R*^2^ = 0.05), while no significant relationship was observed in the L2–L3 context (*b* = −0.009, *t* = −1.55, *p* = 0.127, *R*^2^ = 0.03).

Context comparison revealed that the L1–L3 context showed significantly lower response bias compared to the L1–L2 context (*t* = −3.04, *p* = 0.011, Cohen’s *d* = −0.756), an effect attributable to enhanced LPC amplitudes in the L1–L3 condition.

N400 Amplitudes and Non-Decision Time

N400 amplitudes showed significant negative associations with non-decision time, exclusively in the L2–L3 context (*b* = 0.021, *t* = 3.74, *p* < 0.001, *R*^2^ = 0.21; Figure 11D). No significant relationships emerged in either the L1–L2 context (*b* = 0.0003, *t* = −0.09, *p* = 0.932, *R*^2^ = 0.01) or the L1–L3 context (*b* = 0.007, *t* = 1.32, *p* = 0.190, *R*^2^ = 0.02).

Context comparison revealed that the L2–L3 context exhibited significantly shorter non-decision times compared to the L1–L2 context (*t* = −3.12, *p* = 0.009, Cohen’s *d* = −0.677). This effect is attributable to enhanced N400 amplitudes in the L2–L3 context, indicating increased semantic conflict processing.

## 4. Discussion

### 4.1. Language Comprehension Efficiency Across Dual-Language Contexts

The Adaptive Control Hypothesis (ACH) [72] posits that language control mechanisms adapt to contextual demands. In language production, dual-language contexts are thought to require robust conflict monitoring and costly inhibition of the more proficient language, leading to slower overall processing [30]. While similar context effects have been observed in language comprehension, primarily as a larger Late Positive Component (LPC) in neurophysiological data [31], it remains unclear how processing efficiency varies across different types of dual-language contexts within a single trilingual system. Drawing from models like the Bilingual Interactive Activation (BIA) model, we predicted that larger proficiency gaps (i.e., L1–L3 and L2–L3) would demand greater conflict monitoring and thus exhibit lower processing efficiency compared to the high-proficiency L1–L2 context.

Contrary to our predictions, while raw reaction times showed no difference, a Drift-Diffusion Model (DDM) analysis yielded a surprising outcome: the drift rate—representing the speed of evidence accumulation—was significantly faster in the high-disparity L1–L3 and L2–L3 contexts. This indicates that trilinguals processed information more efficiently when the proficiency gap was larger, a finding that challenges direct applications of production-based control models. We hypothesized that this paradoxical gain stems not from costly inhibition but from a compensatory attentional mechanism that allocates greater bottom-up attention to the less proficient language.

Our neurophysiological data strongly support this attentional account. Compared to the L1–L2 context, the L1–L3 context elicited a larger P300 (indexing attentional capture by novel L3 stimuli) and a smaller N400 (reflecting smoother semantic integration). This P300/N400 pattern diverges from previous findings that identified a larger LPC as the hallmark of dual-language context effects [30,31]. Multivariate Pattern Analysis (MVPA) further revealed that the L1–L3 context was supported by a qualitatively different and sustained neural processing mode. This mode was established early (P200 window) via enhanced cognitive control, leading to more efficient resource utilization in later stages. In the LPC window (822–872 ms), this efficiency was reflected in decreased left temporoparietal activation, while increased right frontal activation, linked to response inhibition, helped participants overcome response bias toward their L1. A temporal generalization analysis confirmed this distinct neural signature was remarkably stable (292–692 ms).

A similar, yet distinct, pattern emerged for the L2–L3 context. While it also showed a larger P300 and smaller N400, MVPA indicated these later-stage effects were primarily quantitative. The crucial qualitative difference was a transient neural pattern in the N2 window (236–322 ms), reflecting heightened conflict monitoring. This early N2 pattern significantly predicted the faster drift rate, a stark contrast to production studies where increased conflict monitoring hinders efficiency [30]. This finding, combined with the absence of an N2-drift rate correlation in the L1–L3 context, supports the ACH by implying that different high-conflict contexts engage distinct underlying mechanisms [73].

Finally, a direct comparison between the L1–L3 and L2–L3 contexts revealed a highly transient (17 ms) qualitative difference in neural patterns during the N2 window, highlighting subtle but distinct adaptations to specific language pairings. This finding points to potential underlying differences that warrant future exploration.

In summary, our findings challenge production-based models of language control by demonstrating that in trilingual comprehension, a greater proficiency gap can paradoxically enhance processing efficiency (a higher drift rate). This gain appears driven by an attentional mechanism (indexed by a P300/N400 signature) rather than costly inhibition. Moreover, we show that different dual-language pairings are managed by qualitatively different, though sometimes transient, neural control mechanisms. While the precise underpinnings of this efficiency gain warrant further investigation, perhaps via time-frequency analysis [74], our study provides compelling evidence for the unique and highly adaptive nature of language control in the trilingual mind.

### 4.2. The Demands of Proactive Control in Different Dual-Language Contexts

The Adaptive Control Hypothesis suggests that proactive control—the pre-emptive adjustment of language activation to prevent interference [5]—is central to bilingualism. In language production, this often involves suppressing the dominant L1, leading to processing costs like a reversed language dominance effect [75] and an enhanced N2 component when switching back to L1 [76]. If this mechanism were identical in comprehension, we would expect a similar N2 effect.

Contrary to production models, our study of trilingual comprehension found no significant N2 amplitude difference between switching to the dominant L1 and the non-dominant L3. Instead, we observed a larger N400 when switching to L1. This suggests that proactive control in comprehension does not involve global language inhibition (an N2-related effect) but rather operates at the semantic level. We attribute this N400 effect to residual interference from L3′s semantic activation impacting L1 integration, a conclusion supported by weight projection analysis showing modulation in the temporal lobe, not the frontal regions typically associated with active inhibition [77,78].

This interference is likely driven by attentional modulation. We observed a smaller P300 when switching to L1, linked to decreased frontal lobe activation, suggesting fewer bottom-up attentional resources were allocated to the familiar L1 [79]. This contrasts with the top-down control seen in production. Essentially, the novel L3 captures more attention, which in turn increases its potential to interfere with the less-attended L1.

MVPA further revealed that proactive control in comprehension is qualitatively different from production. While univariate N2 and LPC amplitudes did not differ between switch conditions, their underlying neural patterns did. This contrasts with production studies, which typically report quantitative amplitude differences with similar neural distributions [80]. The distinct neural patterns we observed—such as reduced right frontal activation when switching to L1—suggest the brain employs fundamentally different strategies for dominant and non-dominant languages rather than merely adjusting the intensity of a single control mechanism.

Overall, the observed neural differences were transient, suggesting that proactive control in trilingual comprehension is weak, aligning with research suggesting such effects are often minimal [12,81]. Our findings diverge from production-based models, indicating that proactive control in comprehension is primarily driven by a bottom-up attentional mechanism. In the L1–L3 context, the dominant L1 is treated as a non-novel stimulus, receives fewer attentional resources, and its semantic processing is subsequently impaired, resulting in a reversal of the language dominance effect [82].

### 4.3. The Demands of Reactive Control in Different Dual-Language Contexts

Research on language control faces a theoretical challenge: the absence of switch costs in comprehension [81], which are robustly found in production. We critically evaluated existing explanations for this discrepancy in trilinguals. Our high-resolution EEG and multivariate pattern analyses revealed no neural differences between switch and repeat trials, refuting the ‘rapid processing’ hypothesis. Furthermore, the consistent presence of a significant N400, a marker of non-target semantic activation [2], contradicts the notion that cross-language interference is too weak to measure.

While the BIA-d model [7] posits that bottom-up comprehension processing involves less inhibition than top-down production [12,21], it fails to account for the complete absence of switch costs we observed, even in the high-conflict L1–L3 context. This suggests the BIA’s language node inhibition mechanism may not be engaged in trilingual comprehension.

We propose an alternative mechanism centered on the LPC. Unlike in production, where the LPC reflects lemma inhibition [80], we found that the LPC did not correlate with switch-cost parameters (i.e., drift rate and non-decision time) [34]. Instead, a larger LPC in the L1–L3 context significantly predicted a reduction in a dominant L1 response bias. We interpret this as reflecting the inhibition of a prepotent response tendency, not non-target words. This process draws on attentional resources, further underscoring the fundamental mechanistic differences between language comprehension and production.

### 4.4. Limitations

While our findings provide evidence for context-specific trilingual control, several limitations should be noted. Methodologically, our modest sample size (*N* = 36) for a complex 12-condition design warrants caution, as techniques like MVPA and DDM are sensitive to individual variability, even though our sample exceeded a priori power analysis requirements. Additionally, the 300 ms stimulus delay, while necessary for avoiding neural contamination, may have limited the interpretation of immediate switching effects. A further limitation concerns the interpretation of our DDM results. While DDM parameters revealed subtle processing differences, their links to our ERP and MVPA findings were not consistently direct, meaning our interpretation remains partly speculative. Future research using single-trial regression analyses could better link these measures. Finally, the generalizability of our results is constrained. Our findings are specific to Uyghur–Chinese–English trilinguals with a unique educational background and may not extend to other multilingual populations. Crucially, the medium-of-instruction relationship between L2 and L3 was a pre-existing characteristic of this group, not an experimentally manipulated variable. Therefore, while we speculate on its role, we cannot draw causal conclusions about its influence on language control. Future research should explicitly test this by comparing different instructional contexts.

## 5. Conclusions

The mechanisms of language switching in trilingual comprehension are significantly different from those in production. First, reactive control is not obligatory in comprehension. Second, proactive control is conditional, emerging only in the L1–L3 context where the proficiency gap is largest. Crucially, the implementation of this proactive control differs from production: instead of inhibiting the dominant language, it involves allocating greater bottom-up attentional resources to the less proficient language (L3) to mitigate cross-linguistic semantic conflict. Finally, unlike in production, this form of proactive control does not hinder overall processing efficiency; on the contrary, it enhances comprehension efficiency in the L1–L3 context.

## Figures and Tables

**Figure 1 brainsci-15-01046-f001:**
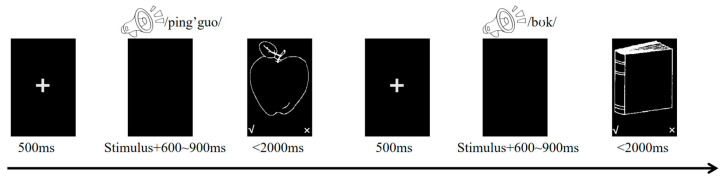
Schematic illustration of the experimental procedure. Participants made word-picture matching judgments by pressing designated response keys. The checkmark (✓) indicates that the auditory word matches the visual picture (match response), while the cross (×) indicates that the auditory word does not match the visual picture (mismatch response). Response key assignment was counterbalanced across participants: half of the participants had the match key (✓) positioned in the lower left and the mismatch key (×) in the lower right, while the other half had the reverse assignment.

**Figure 2 brainsci-15-01046-f002:**
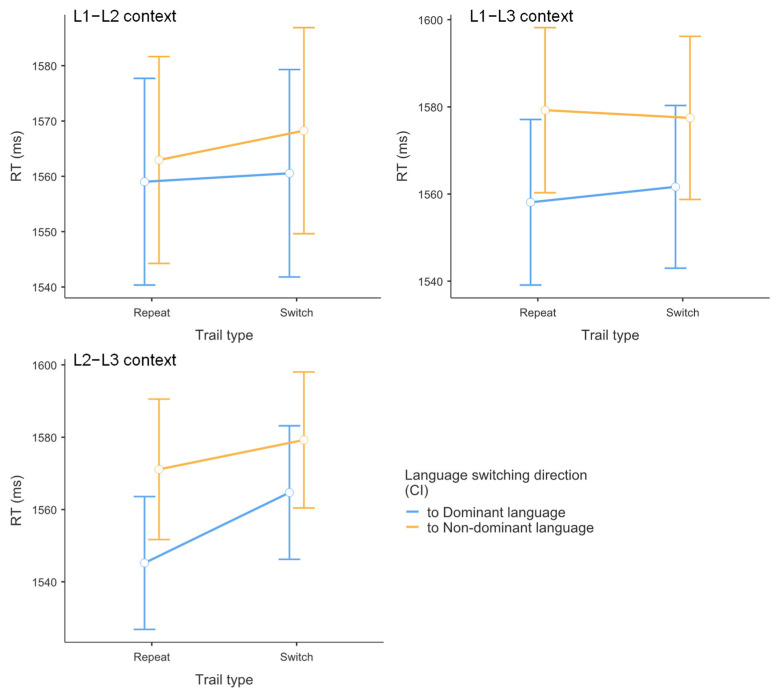
Reaction times (ms) by dual-language contexts, switching direction, and trial type. Error bars represent 95% confidence intervals.

**Figure 3 brainsci-15-01046-f003:**
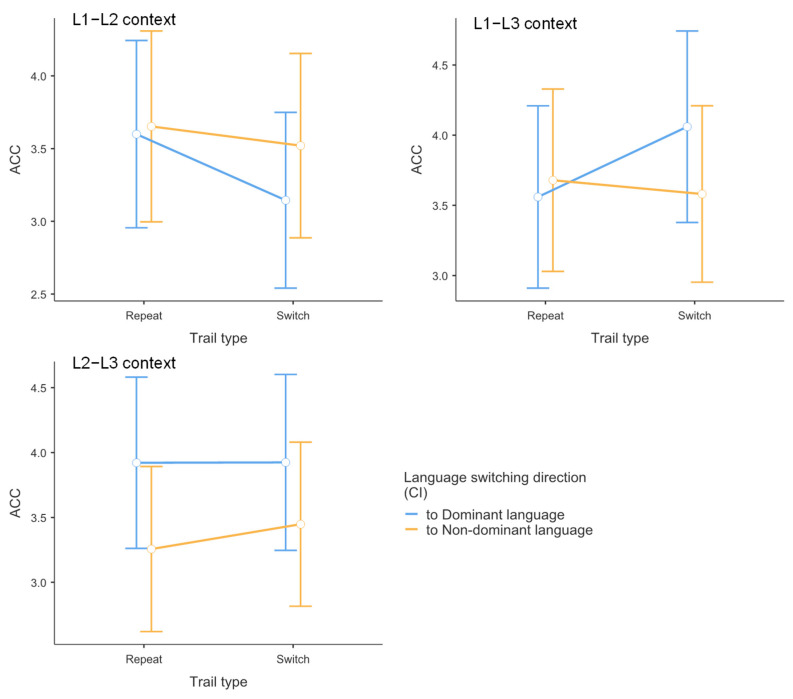
Mean accuracy across dual-language contexts (L1–L2, L1–L3, L2–L3) by switching direction and trial type. Error bars represent 95% confidence intervals.

**Figure 4 brainsci-15-01046-f004:**
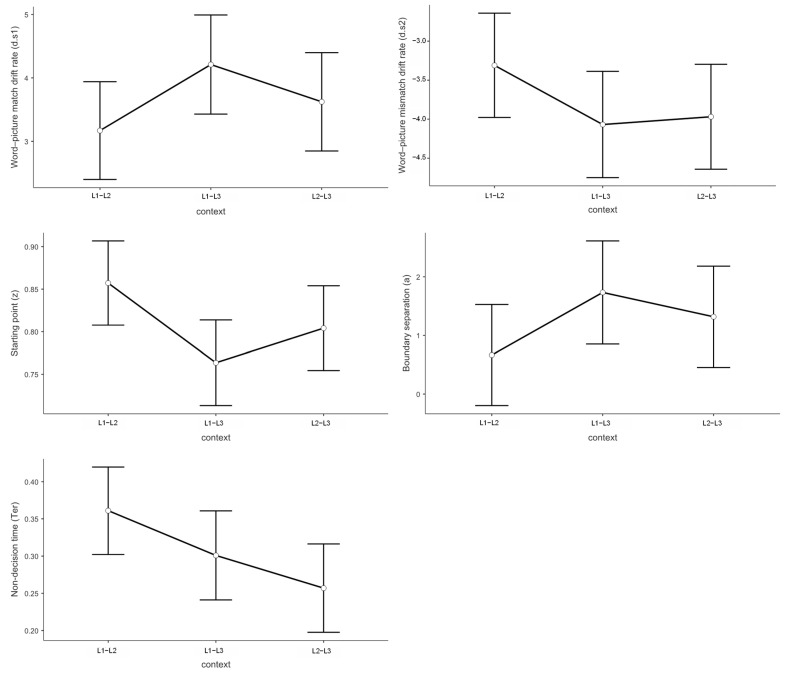
Drift-diffusion model (DDM) parameter estimates for word-picture matching tasks by context. Mean parameter estimates and 95% confidence intervals are shown for drift rate for word-picture matches (**top left**), drift rate for word-picture mismatches (**top right**), starting point (**middle left**), boundary separation (**middle right**), and non-decision time (**bottom**). Data are presented separately for each dual context condition (L1–L2, L1–L3, L2–L3). Error bars represent 95% confidence intervals of the mean (*n* = 36).

**Figure 5 brainsci-15-01046-f005:**
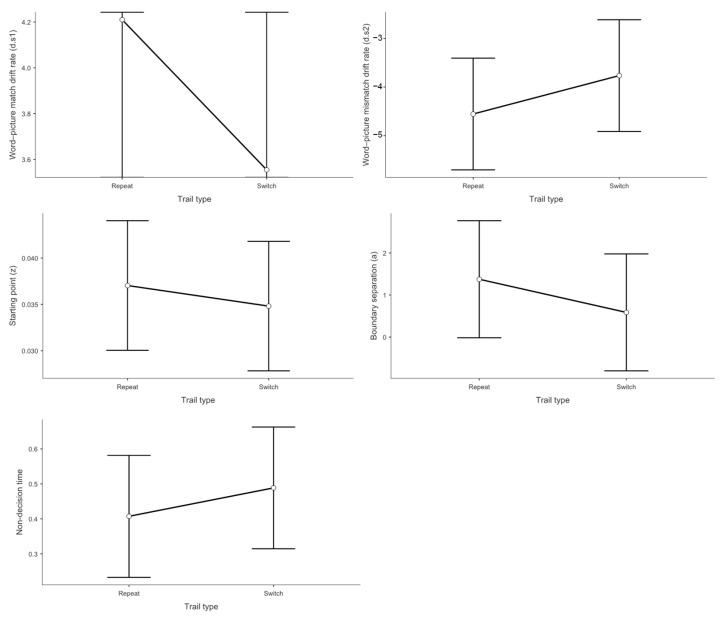
Drift-diffusion model (DDM) parameter estimates for word-picture matching tasks by trial type.

**Figure 6 brainsci-15-01046-f006:**
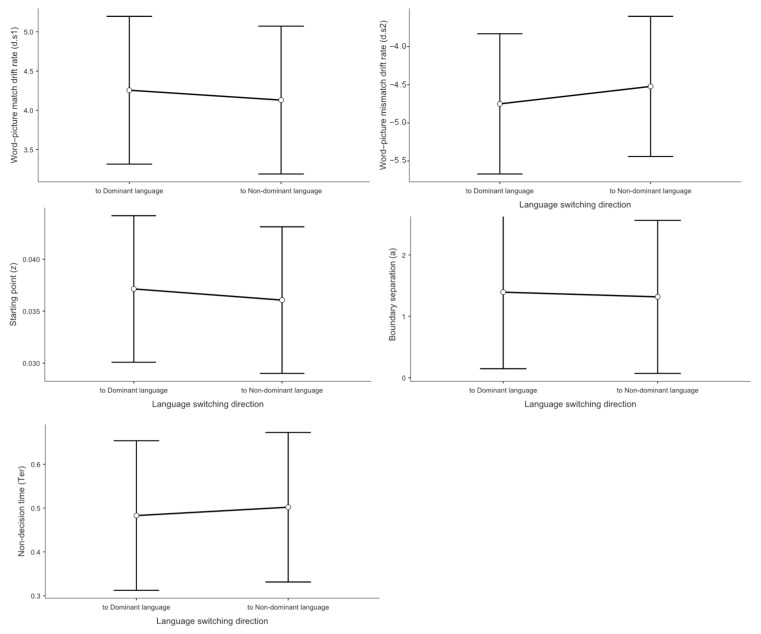
Drift-diffusion model (DDM) parameter estimates for word-picture matching tasks by language switching direction.

**Figure 7 brainsci-15-01046-f007:**
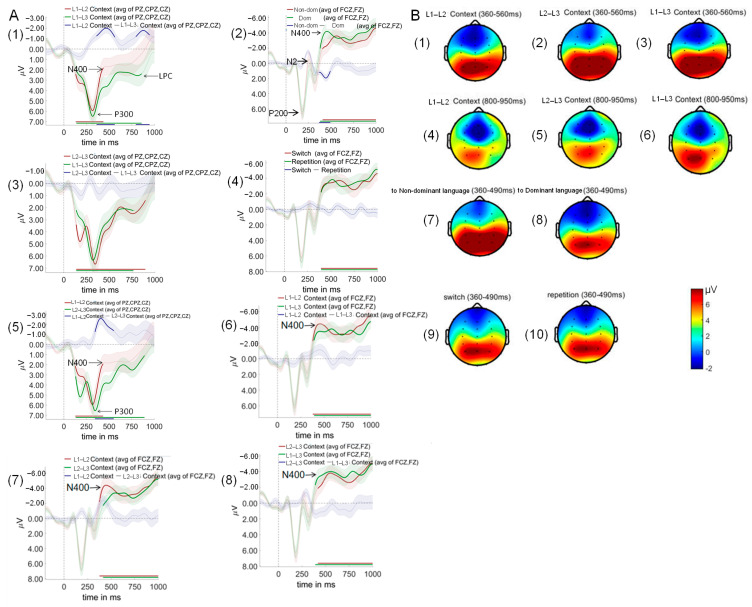
Event-related potentials and scalp topography showing main effects of context, switching direction, and trial type. Panel (**A**) displays grand-averaged event-related potential (ERP) waveforms from representative centro-parietal (PZ, CPZ, CZ) and frontal (FCZ, FZ) electrodes, with time in milliseconds (ms) on the x-axis and voltage in microvolts (μV) on the y-axis. In these plots, red and green traces represent the original waveforms for the compared experimental conditions, while the indigo trace depicts their difference wave; shaded areas denote the standard error of the mean (SEM). Bolded segments on the difference waves indicate significant clusters (*p* < 0.05, identified via cluster-based permutation test). Specifically, at centro-parietal sites, the comparison between L1–L2 and L1–L3 contexts (**A1**) revealed a significant early negative cluster (344–552 ms) and a later positive cluster (790–941 ms), while the L1–L2 vs. L2–L3 context comparison (**A5**) also showed a significant negative cluster (340–547 ms). At frontal sites, switching to a non-dominant (Non-dom) language elicited a significantly greater negativity than switching to a dominant (Dom) language between 369–489 ms (**A2**). Other comparisons for context and trial type (**A3**,**A4**,**A6–A8**) did not yield significant effects. Complementing these temporal dynamics, Panel (**B**) presents scalp topographies of the voltage distribution during key time windows. The topographies for the L1–L2 (**B1**,**B4**), L2–L3 (**B2**,**B5**), and L1–L3 (**B3**,**B6**) contexts are shown for the early (360–560 ms, N400 component) and late (800–950 ms, Late Positive Component) windows. The bottom row illustrates topographies for switching to non-dominant (**B7**) vs. dominant (**B8**) languages, and for switch (**B9**) vs. repetition (**B10**) trials within the 360–490 ms window. For this study, L1, L2, and L3 refer to the participants’ Uyghur, Mandarin Chinese, and English, respectively. General Legend: Red and green traces represent original waveforms for experimental conditions; indigo traces represent difference waves. Shaded areas denote ± standard error of the mean (SEM). Abbreviations: Non-dom, switch to non-dominant language; Dom, switch to dominant language.

**Figure 8 brainsci-15-01046-f008:**
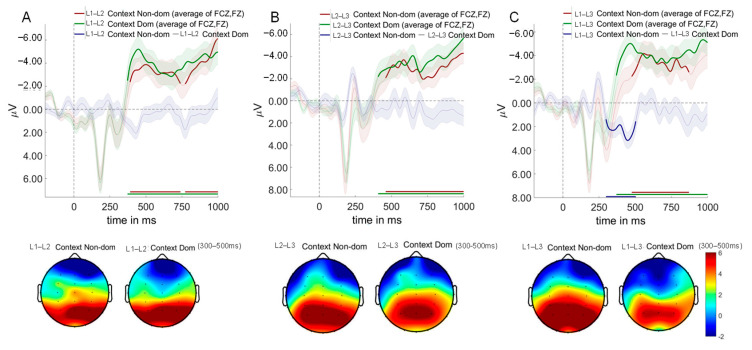
Event-related potentials for language switching directions across dual-language contexts. Red and green traces show original waveforms for switch to non-dominant language and switch to dominant language conditions, respectively; indigo traces show difference waves (switch to non-dominant language minus switch to dominant language). (**A**) L1–L2 context. Upper: ERP waveforms and difference waves. Lower: Scalp topography maps for the 300–500 ms time window. (**B**) L2–L3 context. Upper: ERP waveforms and difference waves. Lower: Scalp topography maps for the 300–500 ms time window. (**C**) L1–L3 context. Upper: ERP waveforms and difference waves. Lower: Scalp topography maps for the statistically significant time window (300–500 ms).

**Figure 9 brainsci-15-01046-f009:**
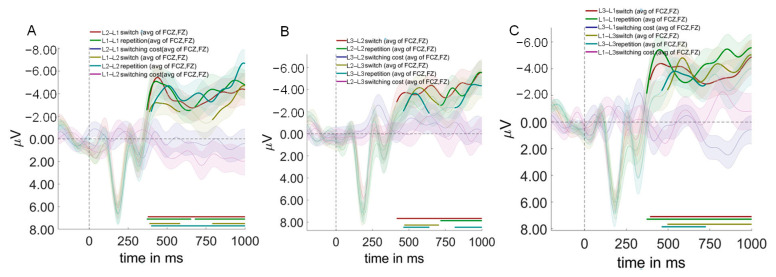
Event-related potentials comparing switch and repeat trials across dual-language contexts and switching directions. Red and yellow traces show switch trial waveforms; green and blue traces show repeat trial waveforms; indigo and purple traces show difference waves (switch minus repeat trials). (**A**) L1–L2 context. ERP waveforms and difference waves for four conditions: switch to dominant language (L2–L1 switch and L1–L1 repeat) and switch to non-dominant language (L1–L2 switch and L2–L2 repeat). (**B**) L2–L3 context. ERP waveforms and difference waves for four conditions: switch to dominant language (L3–L2 switch and L2–L2 repeat) and switch to non-dominant language (L2–L3 switch and L3-L3 repeat). (**C**) L1–L3 context. ERP waveforms and difference waves for four conditions: switch to dominant language (L3–L1 switch and L1-L1 repeat) and switch to non-dominant language (L1–L3 switch and L3–L3 repeat).

**Figure 10 brainsci-15-01046-f010:**
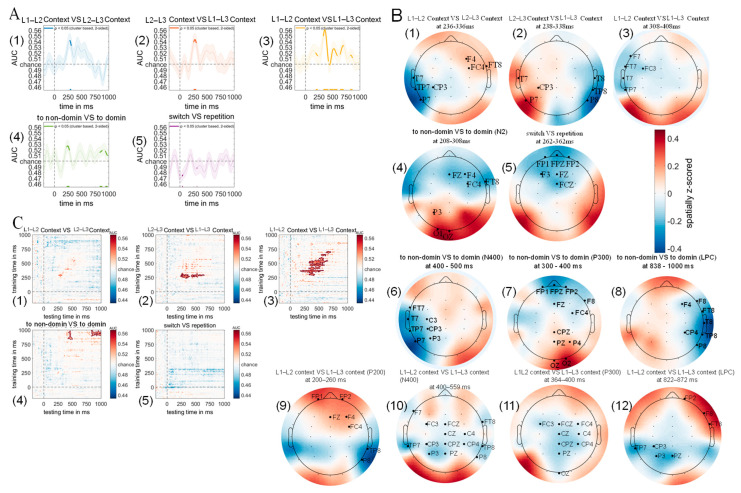
Multivariate pattern analysis (MVPA) results for experimental conditions. (**A**) Diagonal decoding accuracy. This panel presents the time-resolved classification performance (AUC) for five contrasts, reflecting when condition-specific neural patterns were discriminable. Bold lines indicate significant clusters (*p* < 0.05, cluster-based permutation test). (**A1**) The L1–L2 and L2–L3 contexts were dissociable within a 302–329 ms window (peak at 302 ms), corresponding to the N2 component. (**A2**) Similarly, the L1–L3 and L2–L3 contexts were decodable in a 253–299 ms window (peak at 285 ms), also associated with the N2. (**A3**) The L1–L2 and L1–L3 contexts yielded two significant decoding periods: an early window from 182–232 ms (P200 component) and a sustained late window from 330–712 ms (peak at 368 ms), spanning the N2, P300, N400, and LPC components. (**A4**) Switching to the non-dominant versus the dominant language was decodable in three distinct windows: 209–260 ms (peak at 226 ms, N2), 817–861 ms (LPC), and 922–964 ms (LPC). (**A5**) In contrast, no significant time window was found to distinguish switch from repetition trials. (**B**) Activation patterns for key time windows. Spatial distributions of discriminative neural activity (forward models; [62]) are shown for two types of intervals: (1) time points of peak decoding accuracy, representing maximal condition discriminability, and (2) time windows corresponding to significant ERP components (e.g., P200, P300, N400), allowing for functional interpretation of the patterns. Bold electrode markers denote significant clusters (*p* < 0.05). For the context contrasts: (**B1**) Distinguishing L1–L2 from L2–L3 contexts (236–336 ms) was driven by enhanced activity over right fronto-central sites (F4/FC4/FT8) and suppressed activity over left temporo-parietal sites (T7/TP7/CP3/P7). (**B2**) The L1–L3 vs. L2–L3 contrast pattern involved left temporo-parietal enhancement (T7/CP3/P7) and right-hemisphere homologue suppression (T8/TP8/P8). (**B3**) The L1–L2 vs. L1–L3 contrast (308–408 ms) was characterized by widespread suppression over left frontal and temporal sites (F7/FT7/T7/TP8/FC3). (**B9**–**B12**) Further analysis of the L1–L2 vs. L1–L3 contrast revealed distinct topographies for the P200 (**B9**, 200–260 ms: frontal enhancement), P300 (**B11**, 364–400 ms: centro-parietal suppression), N400 (**B10**, 400–559 ms: broad posterior suppression), and LPC (**B12**, 822–872 ms: right frontal enhancement and left posterior suppression) windows. For the switching contrasts: (**B4**) Distinguishing switching directions (208–308 ms) was driven by enhanced activity over left parieto-occipital sites (P3/O1/OZ) and suppression over right frontal sites (FZ/F4/FC4/FT8). (**B5**) The switch vs. repetition contrast (262–362 ms), though not significant in decoding, showed a pattern of frontal suppression (FP1/FPZ/FP2/F3/FZ/FCZ). (**B6**–**B8**) The switching direction contrast also exhibited unique patterns for the N400 (**B6**, 400–500 ms: left temporo-parietal suppression), P300 (**B7**, 300–400 ms: parieto-occipital enhancement and frontal suppression), and LPC (B8, 838–1000 ms: right-hemisphere suppression) windows. (**C**) Cross-temporal generalization matrices. These matrices illustrate the stability of neural representations by training a classifier at one time point (y-axis) and testing it across all others (x-axis). Significant off-diagonal decoding (outlined clusters) indicates that a neural pattern is sustained or re-emerges over time. (**C1**) For the L1–L2 vs. L2–L3 context, decoding was restricted to the diagonal, suggesting a transient neural representation. (**C2**) The L1–L3 vs. L2–L3 context showed transient on-diagonal decoding (266–322 ms) and a significant off-diagonal generalization from the N2 window (training: ~256–307 ms) to the N400 window (testing: ~422–649 ms), suggesting a functional link between early conflict monitoring and later semantic processing. (**C3**) The L1–L2 vs. L1–L3 context demonstrated broad and sustained off-diagonal generalization, indicating a highly stable neural representation. This pattern linked early phonological processing (P200), conflict monitoring (N2), attentional allocation (P300), and semantic processing (N400) (training: ~290–580 ms to testing: ~171–580 ms). Furthermore, distinct stable patterns were observed within the LPC window (e.g., training: ~676–710 ms to testing: ~620–708 ms), suggesting reinstatement of the representation in later stages. (**C4**) For switching direction, a significant off-diagonal generalization was observed from the late LPC window (training: ~826–976 ms) back to the N400 window (testing: ~405–489 ms), implying a functional link between later response inhibition and earlier semantic processing. A stable, sustained pattern was also present within the LPC window itself (training: ~826–976 ms to testing: ~818–982 ms), indicating a persistent representation during this late processing stage. (**C5**) Consistent with the diagonal decoding, no significant off-diagonal generalization was found for switch vs. repetition trials, confirming the lack of a stable, distinguishable neural representation for this contrast. *General Legend*: Shaded areas represent ±SEM. Warm colors (red) in panel C indicate above-chance performance (AUC > 0.5); cool colors (blue) indicate below-chance performance (AUC < 0.5).

**Figure 11 brainsci-15-01046-f011:**
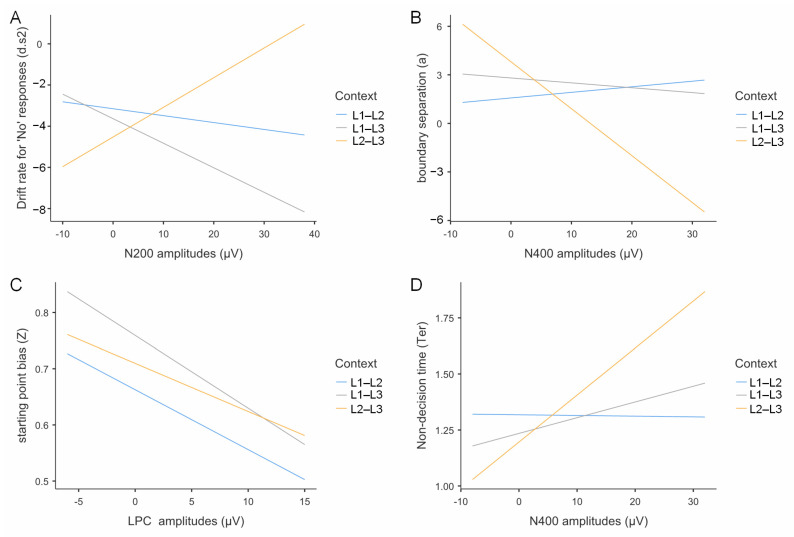
Predictive relationships between ERP components and drift diffusion model parameters across different dual-language contexts. (**A**) Participants’ drift rate for “NO” responses as a function of mean N200 amplitude. Blue lines represent L1–L2 context, gray lines represent L1–L3 context, and orange lines represent L2–L3 context. For the linear regression model, the adjusted response function describes the relationship between the fitted response and N200 amplitudes, with other predictors averaged out by averaging the fitted values over the data used in the fit. Adjusted response data points are computed by adding the residual to the adjusted fitted value for each observation. (**B**) Participants’ boundary separation as a function of mean N400 amplitude. Line colors follow the same convention as Panel A. The adjusted response function describes the relationship between the fitted response and N400 amplitudes, with other predictors averaged out. (**C**) Participants’ starting point bias as a function of mean LPC amplitude. Line colors follow the same convention as Panel A. The adjusted response function describes the relationship between the fitted response and LPC amplitudes, with other predictors averaged out. (**D**) Participants’ non-decision time as a function of mean N400 amplitude. Line colors follow the same convention as Panel A. The adjusted response function describes the relationship between the fitted response and N400 amplitudes, with other predictors averaged out.

**Table 1 brainsci-15-01046-t001:** Participant Demographics.

Characteristic	L1 Uyghur	L2 Chinese	L3 English
Age of acquisition (years) *M* (*SD*)	0.68 (1.80)	5.7 (3.26)	11.87 (3.40)
Exposure (years) *M* (*SD*)	13.46 (5.70)	6.61 (5.41)	0 (0)
Usage (years) *M* (*SD*)	19.02 (2.60)	13.94 (3.27)	8.00 (3.75)
Home Use *M* (*SD*)	2.63 (0.76)	1.45 (0.83)	0.92 (0.68)
Social Use *M* (*SD*)	0.84 (0.79)	3.12 (0.78)	1.04 (0.80)
Self-ratings of proficiency			
Speaking *M* (*SD*)	9.03 (1.24)	8.86 (1.20)	5.48 (1.62)
Listening *M* (*SD*)	9.00 (1.22)	9.10 (1.07)	5.86 (1.20)
Reading *M* (*SD*)	7.65 (2.61)	9.02 (1.11)	6.59 (1.69)
Writing *M* (*SD*)	6.95 (3.21)	9.05 (1.12)	5.97 (1.99)
MINT score *M* (*SD*)	63.0 (1.92)	62.8 (2.10)	49.8 (3.80)
Learning contexts			
Home-only Learning *N* (%)	22 (61.11%)	0 (0%)	0 (0%)
School-only Learning *N* (%)	0 (0%)	30 (83.33%)	36 (100%)
both *N* (%)	14 (38.89%)	6 (16.67%)	0 (0%)
Medium-of-instruction			
Uyghur *N* (%)	n/a	0 (0%)	0 (0%)
Chinese *N* (%)	n/a	36 (100%)	36 (100%)

**Table 2 brainsci-15-01046-t002:** Mean Reaction Times (SE) by Trial Type and Switching Direction Across Dual-Language Contexts.

	L1–L2 Context	L1–L3 Context	L2–L3 Context
to Dominant language			
Repeat *M* (*SE*)	1559 (9.39)	1558 (9.56)	1545 (9.23)
Switch *M* (*SE*)	1561 (9.43)	1562 (9.39)	1565 (9.28)
Switching cost *M* (*SE)*	2 (9.41)	4 (9.47)	20 (9.25)
to Non-dominant language			
Repeat *M* (*SE*)	1563 (9.40)	1579 (9.53)	1571 (9.79)
Switch *M* (*SE*)	1568 (9.36)	1577 (9.41)	1579 (9.46)
Switching cost *M* (*SE*)	5 (9.38)	−2 (9.47)	8 (9.62)
Language dominance effect *M* (*SE*)	11 (9.40)	36 (9.47)	40 (9.44)

*Note*. *SE* = standard error.

**Table 3 brainsci-15-01046-t003:** Mean Accuracy (SE) by Trial Type and Switching Direction Across Dual-Language Contexts.

	L1–L2 Context	L1–L3 Context	L2–L3 Context
to Dominant language			
Repeat *M* (*SE*)	0.973 (0.0085)	0.972 (0.008)	0.981 (0.006)
Switch *M* (*SE*)	0.959 (0.012)	0.983 (0.005)	0.981 (0.006)
Switching cost *M* (*SE)*	−0.014 (0.010)	−0.011 (0.007)	0 (0.006)
to Non-dominant language			
Repeat *M* (*SE*)	0.975 (0.008)	0.975 (0.007)	0.963 (0.011)
Switch *M* (*SE*)	0.971 (0.009)	0.973 (0.008)	0.969 (0.009)
Switching cost *M* (*SE*)	−0.004 (0.008)	−0.002 (0.008)	0.006 (0.010)
Language dominance effect *M* (*SE*)	0.014 (0.009)	−0.007 (0.007)	−0.03 (0.008)

**Table 4 brainsci-15-01046-t004:** Estimated Parameters and *t*-Test Statistics for Each Predictor in the Linear Regression Model.

Outcome	*d.s1*		*d.s2*		a		ter		z	
Predictors	*b*	*t*	*b*	*t*	*b*	*t*	*b*	*t*	*b*	*t*
LPC	−0.023	−0.433, *p* = 0.667	0.0457	1.0543, *p* = 0.295	−0.0606	−1.02, *p* = 0.1	0.01102	1.649, *p* = 0.104	−0.0107	−3.41, *p* = 0.001
Context (L1–L3 vs. L1–L2)	0.958	1.745, *p* = 0.352	−0.9875	−2.05, *p* = 0.047	0.993	1.607, *p* = 0.112	−0.08316	−1.855, *p* = 0.068	0.0923	2.6107, *p* = 0.011
Context (L2–L3 vs. L1–L2)	0.5197	0.937, *p* = 0.085	−1.374	−2.52, *p* = 0.014	1.5658	2.194, *p* = 0.032	−0.12167	−2.4862, *p* = 0.015	0.05	1.4019, *p* = 0.165
Context (L1–L3 vs. L2–L3)	0.4383	0.79, *p* = 0.432	0.891	1.59, *p* = 0.116	0.891	1.59, *p* = 0.116	−0.01189	−0.235, *p* = 0.815	0.01766	−1.964, *p* = 0.685
N200	0.0365	1.03, *p* = 0.304	−0.0154	−0.5, *p* = 0.619	−0.0199	−0.502, *p* = 0.617	0.00376	1.409, *p* = 0.163	−0.0789	−0.0347, *p* = 0.972
N200 × Context (L1–L3 vs. L1–L2)	0.00941	0.1043, *p* = 0.917	−0.0857	−1.147, *p* = 0.256	0.0425	0.435, *p* = 0.665	0.00364	0.539, *p* = 0.592	−0.00163	−0.282, *p* = 0.778
N200 × Context (L2–L3 vs. L1–L2)	−0.10026	−1.101, *p* = 0.275	0.178	2.35, *p* = 0.021	−0.2414	−2.446, *p* = 0.017	0.01108	1.622, *p* = 0.109	−0.00854	−1.461, *p* = 0.149
N200 × Context (L1–L3 vs. L2–L3)	0.10966	0.9944, *p* = 0.324	−0.263	−2.88, *p* = 0.005	0.284	2.376, *p* = 0.02	−0.00744	−0.899, *p* = 0.372	0.0069	0.975, *p* = 0.333
N400	0.0534	1.018, *p* = 0.312	0.0185	0.564, *p* = 0.575	−0.0541	−1.284, *p* = 0.203	0.00627	2.24, *p* = 0.028	0.00419	1.295, *p* = 0.2
N400 × Context (L1–L3 vs. L1–L2)	−0.0389	−0.42, *p* = 0.676	0.00778	0.101, *p* = 0.92	−0.0647	−0.669, *p* = 0.505	0.00732	1.1387, *p* = 0.259	−0.00395	−0.693, *p* = 0.491
N400 × Context (L2–L3 vs. L1–L2)	−0.1157	−1.203, *p* = 0.233	0.2089	2.609, *p* = 0.011	−0.3246	−3.231, *p* = 0.002	0.02127	3.1808, *p* = 0.002	−0.01599	−2.697, *p* = 0.009
N400 × Context (L1–L3 vs. L2–L3)	0.0769	0.6927, *p* = 0.491	−0.201	−2.18, *p* = 0.033	0.26	2.24, *p* = 0.028	−0.0139	−1.809, *p* = 0.075	0.012	1.761, *p* = 0.083

*Note. d.s1*, *d.s2*, *a*, *ter*, and *z* denote drift rate for ‘yes’ responses, drift rate for ‘no’ responses, boundary separation, non-decision time, and starting point bias, respectively. *p* values were corrected for multiple comparisons using the Bonferroni-Holm correction procedure [71].

## Data Availability

The data presented in this study are available on request from the corresponding author, due to privacy restrictions.

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
