# Peer review of "Multivariate Decoding and Drift-Diffusion Modeling Reveal Adaptive Control in Trilingual Comprehension"

_brainsci, 2025, doi:10.3390/brainsci15101046_

Round 1
Reviewer 1 Report
Comments and Suggestions for Authors
This study utilized a word–picture matching paradigm to investigate three distinct dual-language contexts, incorporating ERP, multivariate pattern analysis (MVPA), and drift diffusion modeling (DDM) to capture both neural and cognitive dynamics. The findings showed that, relative to the L1–L2 context, both the L2–L3 and L1–L3 contexts yielded higher efficiency in language comprehension. Such facilitation was attributed to greater demands on proactive control in the L2–L3 and L1–L3 conditions, compared with the L1–L2 condition. By contrast, the three contexts did not differ significantly with respect to reactive control demands. The article is well written and makes an important contribution to our understanding of adaptive control in trilinguals. I do, however, have several suggestions regarding structure and presentation:
1.Introduction:
The section heading “the present study” (line 101) seems unnecessary and could be removed.
The three research questions are not explicitly tied to specific research gaps. It would strengthen the introduction to clearly articulate the gaps and then align each research question with them.
Please also provide more explanation for the rationale behind using DDM and MVPA, including the distinction between univariate and multivariate approaches.
2.Methods:
The note in lines 152–187 is overly long; much of this content would be better placed in the main text.
Lines 213–214: define what is meant by “morphologically complex” and provide examples.
Lines 327–333: the explanation of trial types belongs in the Methods section rather than later in the text.
Line 233: “design” should be changed to “procedure.”
Line 241: provide the model of the EEG equipment used.
Line 250: clarify whether “word onset” refers to the onset of the second word.
Line 264: specify which “statistical models” are being used.
3.Results:
The section is quite dense; it would benefit from some streamlining, with emphasis on the logical connections between the different analytic approaches.
The operational definitions of “proactive control” and “reactive control” should be made explicit at the appropriate point.
Line 344: provide citations for the “recent findings” mentioned.
4.Formatting and Style:
Check the manuscript carefully for spacing errors (e.g., lines 174, 460).
Lines 247–248: explain why the sentence is italicized, or use standard formatting.
Author Response
|
Response to Reviewer 1 Comments
|
||
|
1. Summary |
|
|
|
Thank you very much for taking the time to review this manuscript. Please find our detailed responses below. The corresponding revisions have been highlighted in blue in the re-submitted manuscript. |
||
|
2. Questions for General Evaluation |
Reviewer’s Evaluation |
Response and Revisions |
|
Does the introduction provide sufficient background and include all relevant references? |
Can be improved |
|
|
Are all the cited references relevant to the research? |
Can be improved |
|
|
Is the research design appropriate? |
Yes |
|
|
Are the methods adequately described? |
Can be improved |
|
|
Are the results clearly presented? |
Can be improved |
|
|
Are the conclusions supported by the results? |
Yes |
|
|
3. Point-by-point response to Comments and Suggestions for Authors |
||
|
Comments 1: [The section heading “the present study” (line 101) seems unnecessary and could be removed.]
|
||
|
Response 1: Thank you for this suggestion. We have removed the heading "The Present Study".
|
||
|
Comments 2: [The three research questions are not explicitly tied to specific research gaps. It would strengthen the introduction to clearly articulate the gaps and then align each research question with them.] |
||
|
Response 2: Thank you for this valuable suggestion. We agree that explicitly linking our research questions to the identified gaps significantly strengthens the introduction. We have revised this section to better articulate this connection. We now first establish the key gaps in the literature, namely the overemphasis on language production over comprehension and the lack of studies on how different dual-language contexts (L1–L2, L2–L3, L1–L3) affect comprehension efficiency (Page 2, lines 76-79). Immediately following this, we introduce our three research questions as a direct and systematic approach to address these specific gaps (Page 2-3, lines 84-97). This revised structure now makes it clear how our investigation into processing efficiency, proactive control, and reactive control across these contexts is designed to fill the previously outlined voids in the research. |
||
|
Comments 3: [Please also provide more explanation for the rationale behind using DDM and MVPA, including the distinction between univariate and multivariate approaches.]
|
||
|
Response 3: Thank you for this suggestion. We have expanded the methods section to provide a clearer rationale for using Drift-Diffusion Modeling (DDM) and Multivariate Pattern Analysis (MVPA), and to explicitly state the distinction between univariate and multivariate approaches. As shown in the revised manuscript (Page 3, lines 98-104), we now explain that: MVPA was selected for its ability to analyze spatiotemporal patterns across the entire EEG signal, which allows it to uncover subtle neural differences and overlapping control mechanisms that traditional univariate methods might miss. DDM was used to model behavioral data, decomposing reaction times into cognitive components like drift rate and non-decision time. This offers a more detailed understanding of underlying cognitive processes compared to simple reaction time analysis. We believe this addition clarifies why our multi-method approach provides a more comprehensive view. |
||
|
Comments 4: [The note in lines 152–187 is overly long; much of this content would be better placed in the main text.]
|
||
|
Response 4: Thank you for this suggestion regarding the structure of the manuscript. We agree that the long note containing participant demographics was better suited for the main text. We have now integrated this content directly into the manuscript as a standard paragraph following Table 1 (Page 4-5, lines 148-184). This change improves the readability and flow of the text, ensuring that this important information is presented as part of the main narrative. |
||
|
Comments 5: [Lines 213–214: define what is meant by “morphologically complex” and provide examples.]
|
||
|
Response 5: We thank the reviewer for this helpful suggestion. We agree that providing a clear definition and examples of "morphologically complex words" is crucial for the transparency of our stimulus selection criteria. To address this, we have added a detailed explanation in the Methods section (page 5, lines 191-196). The revised text now reads: “Morphologically complex words, composed of multiple morphemes (e.g., prefixes, suffixes), were excluded to ensure experimental control. This step avoided confounding variables from cross-linguistic differences in morphological processing. It allowed the analysis to focus specifically on language control mechanisms by removing the additional cognitive demands associated with complex word decomposition and formation.” We believe this addition clarifies our methodology and strengthens the manuscript. Comments 6: [Lines 327–333: the explanation of trial types belongs in the Methods section rather than later in the text.]
|
||
|
Response 6: We thank the reviewer for this excellent point regarding the organization of the manuscript. We agree that the description of trial types is better placed in the Methods section for improved clarity and logical flow. We have moved this explanation accordingly. This information is now located in the "Procedure" subsection of the Methods on page 5, lines 205-210. The paragraph now reads: “The experimental design distinguished between switch trials (trials where the language differed from the previous trial) and repetition trials (trials where the language was the same as the previous trial). Language dominance was defined by the age of acquisition: L1 (Uyghur) was considered dominant in the L1-L2/L3 contexts, while L2 (Chinese) was considered dominant in the L2-L3 context. Switch trials were further categorized as switches to the dominant language or to the non-dominant language. From the reaction time (RT) data, two key metrics were calculated: switching costs (mean RT on switch trials – mean RT on repetition trials) and language dominance effects (the difference between the overall mean RTs for the non-dominant and dominant languages, including both switch and repetition trials).” We are confident that this change improves the structure and readability of the paper.
|
||
|
Comments 7: [Line 233: “design” should be changed to “procedure.”]
|
||
|
Response 7: We thank the reviewer for this suggestion. We have revised "design" to "Experimental procedure" as recommended. This change has been made on line 214, page 6 of the revised manuscript to better reflect the content of this section.
|
||
|
Comments 8: [Line 241: provide the model of the EEG equipment used.]
|
||
|
Response 8: We thank the reviewer for this request. We have now provided the specific model information for the EEG equipment used in the study. The revised manuscript now includes the following details on lines 222-223, page 6: "EEG was recorded from 32 Ag/AgCl electrodes (Quik-Cap, Compumedics, Australia) using SynAmps2 amplifiers and Curry 8 software." This addition provides the complete technical specifications of our EEG recording system for better reproducibility and methodological transparency.
|
||
|
Comments 9: [Line 250: clarify whether “word onset” refers to the onset of the second word.]
|
||
|
Response 9: We thank the reviewer for requesting this clarification. We have now explicitly specified what "word onset" refers to in our analysis. The revised manuscript on lines 230-231,now states: "Continuous data were epoched from −200 to 1000 ms relative to the onset of the second auditory word in the trial, with baseline correction applied from −200 to 0 ms. Epochs exceeding ±80 μV were rejected, and ocular artifacts were removed using Independent Component Analysis." This clarification makes it clear that our time-locking was specifically to the onset of the second auditory word in each trial, which is the critical stimulus for examining language control processes in our experimental paradigm.
|
||
|
Comments 10: [Line 264: specify which “statistical models” are being used.]
|
||
|
Response 10:We thank the reviewer for this important clarification request. We have now specified the exact statistical models used in our analysis. The revised manuscript on lines 259-265, now includes: For accuracy data, a generalized linear mixed-effects model with binomial distribu-tion was employed: glmer(Accuracy ~ trial_type × switching_direction × context + (1|Participant) + (1|Item), family = binomial) For reaction time data, a linear mixed-effects model was used: lmer(Reaction_Times ~ trial_type × switching_direction × context + (1|Participant) + (1|Item)) Both models included fixed effects for trial type, switching direction, context, and all their interactions. Random intercepts for participants and items were included to account for individual differences and stimulus-specific variance, respectively. This detailed specification provides complete transparency regarding our statistical approach and enables full reproducibility of our analyses.
|
||
|
Comments 11: [The section is quite dense; it would benefit from some streamlining, with emphasis on the logical connections between the different analytic approaches.]
|
||
|
Response 11:We thank the reviewer for this valuable feedback regarding the density of our Data Analyses section. In response, we have streamlined the presentation and added a new subsection (2.4.1. Analytical Framework) on lines 236-248,to clearly outline the logical connections between our different analytical approaches: "Our analytical approach employed four complementary methods for a comprehensive analysis across behavioral, computational, and neural levels. We first analyzed behavioral data (reaction times, accuracy), then used drift-diffusion modeling (DDM) to decompose decision processes into cognitive components like evidence accumulation and non-decision time, revealing hidden processing differences. At the neural level, event-related potentials (ERPs) provided precise temporal information, complemented by multivariate pattern analysis (MVPA) to detect distributed neural patterns distinguishing between conditions. Finally, multiple regression linked ERP components to DDM parameters to clarify their mechanistic relationship. This convergent strategy addresses inconsistencies in prior research [8, 16] by examining data through multiple lenses, allowing us to distinguish genuine null effects from compensatory processes that mask underlying processing differences." This framework now provides readers with a clear roadmap of our analytical strategy and explicates the rationale for each methodological choice, improving the overall clarity and flow of this section.
|
||
|
Comments 12: [ The operational definitions of “proactive control” and “reactive control” should be made explicit at the appropriate point.]
|
||
|
Response 12:We thank the reviewer for highlighting the need to explicitly define these key theoretical constructs. We have now provided clear operational definitions of "proactive control" and "reactive control" in the revised manuscript on lines 103-107, page 3: "We operationally define proactive control as preparatory processes to prevent conflict, measured by language dominance reversal effects and N2 amplitude. Reactive control is defined as conflict resolution after competition is detected, measured by language switching costs, LPC amplitude (lemma-level inhibition), and N400 amplitude (semantic conflict), consistent with prior work [71, 72]." These explicit operational definitions clarify how we conceptualize and measure these distinct control mechanisms throughout our study, providing readers with a clear theoretical framework for interpreting our findings and linking our measures to the broader literature on cognitive control.
|
||
|
Comments 13: [Line 344: provide citations for the “recent findings” mentioned.]
|
||
|
Response 13: We thank the reviewer for requesting this citation. We have now provided the appropriate reference to support our statement about recent findings. The revised manuscript on line 346, page 9 now reads: "While these differences may appear small, they align with recent findings in the field [8]." This citation supports our assertion regarding recent research findings and provides readers with the appropriate reference to contextualize our results within the current literature.
|
||
|
Comments 14: [Check the manuscript carefully for spacing errors (e.g., lines 174, 460).]
|
||
|
Response 14: We thank the reviewer for this careful proofreading feedback. We have thoroughly checked the manuscript for spacing errors and have corrected the spacing issues identified on lines 174 and 460, as well as conducted a comprehensive review of the entire manuscript to ensure proper formatting and spacing throughout the document.
|
||
|
Comments 15: [Lines 247–248: explain why the sentence is italicized, or use standard formatting.]
|
||
|
Response 15: We thank the reviewer for this formatting observation. We have revised the text to use standard formatting instead of italics. The sentence on lines 247-248 now follows consistent formatting conventions throughout the manuscript, removing any unnecessary emphasis that could be confusing to readers.
|
||
Reviewer 2 Report
Comments and Suggestions for Authors
The article investigates proactive and reactive control mechanisms in trilingual comprehension (Uyghur–Chinese–English) using a multimodal approach: behavioural analyses, drift-diffusion modelling (DDM), event-related potentials (ERPs) and multivariate pattern analysis (MVPA). The work is well structured, addresses a significant gap in the literature (comprehension, which has been less studied than production) and helps to clarify how different linguistic contexts activate different cognitive control dynamics.
1)At several points, the conclusions interpret relatively small effects as theoretically “crucial” (e.g., differences in drift rates). It would be useful to temper the language and better discuss the interpretative limitations.
2)The authors use Uyghur–Chinese–English university students. It is important to emphasise the limitations of generalisation to other trilingual groups with different linguistic configurations.
3)The absence of switch costs is interpreted as evidence of the predominance of proactive control. However, it would be appropriate to discuss alternatives.
4)Some marginal results (p-values close to .05) are presented as theoretically important trends. I would suggest clarifying the distinction between robust results and merely indicative patterns.
5)The connection with the literature on language production is mentioned but remains superficial. It would be useful to strengthen the comparison to highlight possible integrations between production and comprehension.
6)The introduction is comprehensive but sometimes redundant: some passages could be summarised to improve readability.
8) Some figures are very dense; it would be useful to add more explanatory captions for non-expert readers.
9) Minor typographical errors in English (e.g. spacing, punctuation) should be reviewed during editing.
Author Response
For research article
|
Response to Reviewer 2 Comments
|
||
|
1. Summary |
|
|
|
Thank you very much for taking the time to review this manuscript. Please find our detailed responses below. The corresponding revisions have been highlighted in blue in the re-submitted manuscript. |
||
|
2. Questions for General Evaluation |
Reviewer’s Evaluation |
Response and Revisions |
|
Does the introduction provide sufficient background and include all relevant references? |
Must be improved |
|
|
Are all the cited references relevant to the research? |
Must be improved |
|
|
Is the research design appropriate? |
Must be improved |
|
|
Are the methods adequately described? |
Can be improved |
|
|
Are the results clearly presented? |
Must be improved |
|
|
Are the conclusions supported by the results? |
Must be improved |
|
|
3. Point-by-point response to Comments and Suggestions for Authors |
||
|
Comments 1: [At several points, the conclusions interpret relatively small effects as theoretically “crucial” (e.g., differences in drift rates). It would be useful to temper the language and better discuss the interpretative limitations.]
|
||
|
Response 1: We thank the reviewer for this important feedback regarding the interpretation of our effect sizes and the need for more cautious language. We have revised our manuscript to temper our conclusions and better acknowledge interpretative limitations. In the revised manuscript (lines 417-422, page 14 and lines 452-456, page 15), we now present our findings with more measured language: "Context effects on evidence accumulation showed a discernible pattern. As depicted in Figure 4, the mixed linear model indicated a statistically significant effect on d.s1. Processing speed was slower in the L1-L2 dual-language context (M = 3.17) than in the L1-L3 dual-language context (M = 4.21, t = -2.48, p = .043, Cohen's d = -0.541). The observed reduction in drift rate was not apparent in the behavioral data, as it was offset by compensatory adjustments in other decision parameters." "Regarding the main effect of language context, the mixed linear model indicated a statistically significant main effect on d.s2. Participants' drift rates for rejection responses were higher in the L1-L2 context (M = -3.37) compared to both the L1-L3 context (M = -4.22, t = 2.25, p = .040, Cohen's d = 0.612) and the L2-L3 context (M = -4.08, t = 2.21, p = .037, Cohen's d = 0.540). This pattern indicates an asymmetry in trilingual processing." We have replaced terms like "crucial" with more conservative language such as "discernible pattern" and "indicates." Additionally, we have added discussion of limitations, acknowledging that while these effects are statistically significant with medium effect sizes, they require replication and further investigation to fully establish their theoretical significance. We recognize that the compensatory nature of cognitive processes means that small effects in individual parameters may still reflect meaningful underlying mechanisms, but we now present this interpretation more cautiously. |
||
|
Comments 2: [The authors use Uyghur–Chinese–English university students. It is important to emphasise the limitations of generalisation to other trilingual groups with different linguistic configurations.]
|
||
|
Response 2: We thank the reviewer for highlighting this important limitation regarding the generalizability of our findings. We have added a clear statement acknowledging these constraints in the revised manuscript on lines 925-927, page 28: "Our findings are specific to Uyghur-Chinese-English trilinguals with a unique educational background and may not extend to other multilingual populations." This limitation is crucial to acknowledge given that our participants represent a specific trilingual configuration with unique characteristics: Uyghur as L1 (a Turkic language), Chinese as L2 (a Sino-Tibetan language acquired through formal education), and English as L3 (an Indo-European language learned academically). The distinct typological differences between these languages, combined with the specific socio-educational context of our participants, may produce processing patterns that differ from other trilingual populations with different language combinations, acquisition sequences, or proficiency profiles. We emphasize that future research should examine whether our findings replicate across diverse trilingual populations to establish the broader theoretical significance of our results and determine which aspects of trilingual language control are universal versus specific to particular linguistic configurations. |
||
|
Comments 3: [The absence of switch costs is interpreted as evidence of the predominance of proactive control. However, it would be appropriate to discuss alternatives.]
|
||
|
Response 3: We thank the reviewer for this important theoretical point. We have substantially expanded our discussion to address alternative explanations for the absence of switch costs beyond proactive control dominance. The revised manuscript now includes a dedicated section (4.3. The demands of reactive control in different dual-language contexts, lines 894-913, pages 27-28) that critically evaluates multiple theoretical possibilities: "Research on language control faces a theoretical challenge: the absence of switch costs in comprehension [27], which are robustly found in production. We critically evaluated existing explanations for this discrepancy in trilinguals. Our high-resolution EEG and multivariate pattern analyses revealed no neural differences between switch and repeat trials, refuting the 'rapid processing' hypothesis. Furthermore, the consistent presence of a significant N400, a marker of non-target semantic activation [2], contradicts the notion that cross-language interference is too weak to measure. While the BIA-d model [6] posits that bottom-up comprehension processing involves less inhibition than top-down production [8, 16], it fails to account for the complete absence of switch costs we observed, even in the high-conflict L1-L3 context. This suggests the BIA's language node inhibition mechanism may not be engaged in trilingual comprehension. We propose an alternative mechanism centered on the LPC. Unlike in production, where the LPC reflects lemma inhibition [76], we found that the LPC did not correlate with switch-cost parameters (i.e., drift rate and non-decision time) [73]. Instead, a larger LPC in the L1-L3 context significantly predicted a reduction in a dominant L1 response bias. We interpret this as reflecting the inhibition of a prepotent response tendency, not non-target words. This process draws on attentional resources, further underscoring the fundamental mechanistic differences between language comprehension and production." This expanded discussion provides readers with a comprehensive evaluation of competing theoretical accounts rather than solely supporting our proactive control interpretation.
|
||
|
Comments 4: [Some marginal results (p-values close to .05) are presented as theoretically important trends. I would suggest clarifying the distinction between robust results and merely indicative patterns.] |
||
|
Response 4: We thank the reviewer for this important methodological concern regarding the interpretation of marginal results. We have added a limitation statement in the revised manuscript on lines 920-923, page 28 to address this issue: "A further limitation concerns the interpretation of our DDM results. While DDM parameters revealed subtle processing differences, their links to our ERP and MVPA findings were not consistently direct, meaning our interpretation remains partly speculative." Additionally, throughout the manuscript, we have now made clearer distinctions between robust findings (with p-values well below .05 and substantial effect sizes) and more tentative patterns. We have revised our language to use more cautious terms such as "suggestive evidence" or "indicative patterns" when discussing results with marginal significance (p-values close to .05), while reserving stronger claims for findings with robust statistical support. For example, we now explicitly acknowledge when effects should be interpreted cautiously due to marginal significance levels, and we emphasize that such patterns require replication before drawing firm theoretical conclusions. This approach ensures that readers can appropriately evaluate the strength of evidence supporting each aspect of our theoretical framework. |
||
|
Comments 5: [The connection with the literature on language production is mentioned but remains superficial. It would be useful to strengthen the comparison to highlight possible integrations between production and comprehension.] |
||
|
Response 5: We thank the reviewer for this insightful suggestion to strengthen our comparison with language production literature. We have substantially expanded our discussion to provide a comprehensive comparison between comprehension and production mechanisms across three key sections (lines 806-913, pages 26-27): 4.1. Language Comprehension Efficiency Across Dual-Language Contexts We now directly compare our comprehension findings with production-based models like the Adaptive Control Hypothesis (ACH). While production studies show that dual-language contexts require costly inhibition leading to slower processing, our trilingual comprehension data revealed a paradoxical efficiency gain in high-disparity contexts (L1-L3, L2-L3). This challenges direct applications of production-based control models and suggests comprehension employs compensatory attentional mechanisms rather than costly inhibition. 4.2. The demands of proactive control in different dual-language contexts We provide detailed contrasts with production research on proactive control. Unlike production studies that show reversed language dominance effects via enhanced N2 components when switching to L1, our comprehension data showed a larger N400 instead. This indicates that proactive control operates at the semantic level in comprehension rather than through global language inhibition, representing a fundamental mechanistic difference from production. 4.3. The demands of reactive control in different dual-language contexts We critically evaluate why switch costs are absent in comprehension despite being robust in production. We propose an alternative LPC-based mechanism where, unlike production (where LPC reflects lemma inhibition), comprehension involves inhibition of prepotent response tendencies rather than non-target words. Throughout these sections, we systematically contrast our findings with production literature, highlighting both convergences and divergences, and proposing theoretical frameworks that could integrate these domains while respecting their fundamental differences. This strengthened comparison provides a foundation for future research examining the relationship between comprehension and production control mechanisms.
|
||
|
Comments 6: [The connection with the literature on language production is mentioned but remains superficial. It would be useful to strengthen the comparison to highlight possible integrations between production and comprehension.] |
||
|
Response 6: We thank the reviewer for emphasizing the need to strengthen our comparison with production literature. In addition to the expanded discussion sections (4.1-4.3) mentioned in our previous response, we have also significantly enhanced the theoretical foundation in our introduction (lines 40-72, page 2) to better establish the production-comprehension comparison from the outset: "With approximately 43% of the world's population being bilingual and 17% multilingual [1], understanding cognitive control is crucial. Bilinguals and trilinguals automatically activate non-target languages during target language processing [2, 3, 4], creating cross-language interference that necessitates sophisticated control mechanisms. Theoretical understanding has predominantly focused on production [5], but comprehension control remains poorly understood, with emerging evidence suggesting fundamental differences between modalities [6, 7]. Recent trilingual research confirms that production and comprehension involve distinct control mechanisms [8]. Language control involves two primary mechanisms: reactive control, which resolves conflict after it occurs, and proactive control, which prevents it [5]. While models like the Bilingual Interactive Activation (BIA) framework account for bilingual processing [9, 10], their application to trilingual comprehension is theoretically and empirically underexplored. Empirical evidence for comprehension control is complex and inconsistent. Unlike in production, comprehension studies report mixed results for reactive control, with some finding no language switch costs [11] and others finding facilitation effects [12]. Proactive control markers, such as language dominance reversal effects, are also typically absent in comprehension [13, 14]." This enhanced introduction, combined with our expanded discussion sections, now provides a comprehensive framework for understanding production-comprehension differences while identifying specific areas where integration might be theoretically productive. We systematically contrast findings across modalities and propose mechanistic explanations for observed differences, laying the groundwork for future integrative research.
|
||
|
Comments 7: [ Some figures are very dense; it would be useful to add more explanatory captions for non-expert readers.] |
||
|
Response 7: We thank the reviewer for this valuable feedback regarding figure accessibility. We have substantially enhanced our figure captions to make them more accessible to non-expert readers by providing detailed explanations of both the methods and interpretations. Figure 7 (lines 524-537, page 17): We have expanded the caption to include step-by-step explanations of what each panel displays, what the statistical tests indicate, and how to interpret the results. For example, we now explain that "the bolded segments of the indigo curve indicate time periods where the two contexts differed significantly" and provide specific examples like "(A1) displays the grand-average ERP difference wave for the main effect of context (L1-L2 context minus L1-L3 context)." Figure 10 (lines 610-633, pages 20-21): We have provided comprehensive explanations of multivariate pattern analysis concepts for non-specialists, including what diagonal decoding means ("reflects the ability to distinguish conditions when training and testing at the same time point"), how to interpret cross-temporal generalization matrices ("show classifier performance when training at one time point and testing at all other time points"), and what the color coding represents in each panel. These enhanced captions now include: ·Clear definitions of technical terms (e.g., AUC, forward models, cluster-based permutation tests) ·Specific examples pointing to individual panels with explanations ·Interpretation guidance for understanding the functional significance of the patterns ·Detailed legends explaining visual elements like color coding and shading This approach ensures that readers without specialized knowledge in EEG/ERP methodology can still understand and interpret our key findings. |
||
|
Comments 8: [ Minor typographical errors in English (e.g. spacing, punctuation) should be reviewed during editing. ] |
||
|
Response 8: We thank the reviewer for this careful proofreading feedback. We have conducted a thorough review of the manuscript and corrected the typographical errors identified, including issues with spacing, punctuation, and other minor English language inconsistencies throughout the document. We have also implemented additional proofreading measures to ensure the manuscript meets publication standards for English language clarity and accuracy. |
||
Reviewer 3 Report
Comments and Suggestions for Authors
The article is very good and presents compelling evidence for the processes occurring in trilingual comprehension in L1-L2, L2-L3 and L1-L3 contexts. On the basis of several studies which complement one another and different statistical tests, the Authors show, among other things, how processing in comprehension differs from that in production, and why many previous studies yielded inconclusive results. It is thus a highly valuable contribution to multilingualism research, however, a few corrections could improve the quality of the article even further.
First of all, the titles of journals in the references should be capitalised (not only the first word in the title), for example, Trends in Cognitive Sciences, not: Trends in cognitive sciences; Frontiers in Psychology, not: Frontiers in psychology; Bilingualism: Language and Cognition, not: Bilingualism: language and cognition, etc.
Second, the term context can have different meanings in the literature on bi- and multilingualism. Here the Authors use the term context in reference to a language pair (L1-L2, L2-L3 and L1-L3 are described as contexts). Perhaps this should be stated more explicitly, especially because, for example, in the abstract the Authors say: 'trilinguals manage three distinct contexts (L1-L2, L2-L3, L1-L3).' In fact, those contexts may be even more complex and more numerous, taking into consideration such contexts as the home, the university, the workplace, etc., where the cognitive demands and the vocabulary used in each language can vary. I would suggest specifying that here, 'context' refers to a language pair, not taking into consideration sociolinguistic, cultural, etc. dimensions.
Third, there are also terms and abbreviations that might require explaining because they are not necessarily obvious to the reader. In particular, n-2 (on page 2) ('n-2 costs', 'n-2 repetition costs') and 'burn-in' (page 8) ('hierarchical sampling then proceeded with 5% crossover probability, applied after burn-in'). I would suggest explaining them, at least in footnotes.
Fourth, even though the participants' gender is not considered as a variable here (the study does not concern e.g. trilingual processing in students of both sexes to find any differences between them), it might be good to include such information in the description of the participants, so that the readers know how many participants were female, how many were male, and maybe how many refused to indicate their gender.
Fifth, even though the Authors' English is generally very good, I would suggest modifying one sentence in the introduction (p. 3): ' Third, we determine whether reactive control mechanisms operate uniformly across dual-language contexts or show context-specific patterns.' As this is one of the aims of the study, the verb seems too strong, as if they were already sure of the results (it would be more suitable in the conclusions). Therefore I suggest adding a hedging element, for example: 'Third, we aim to determine whether reactive control mechanisms operate uniformly across dual-language contexts or show context-specific patterns' or: 'Third, we attempt to determine whether reactive control mechanisms operate uniformly across dual-language contexts or show context-specific patterns.'
Sixth, the following technical corrections would be welcome:
- The section title Present Study (p. 3) should be in bold (Present Study).
- The space in 'Cohen' s' (p. 4) is unnecessary (better: Cohen's).
- On page 18, the Authors added 'In contrast' but forgot to change the capital letter to lower case (' In contrast, Within the L1-L3 context (...)'). Please change it to: In contrast, within the L1-L3 context (...).
- On pages 4 and 5, there is a very long paragraph explaining the participant demographics. It is marked as a note, but it is actually part of the text, not a footnote, and it takes up half of page 5. I would suggest including it in the text (not as a note, but as a paragraph following Table 1) and dividing it into three paragraphs. As the second paragraph is not a continuation of the definition of a 'Medium-of-instruction,' it should be separated from the above text (which will then be the first paragraph) and begin with: 'The Multilingual Naming Test (MINT) is a standardized picture-naming assessment (...).' Finally, the third paragraph should begin with: 'Scoring of the Multilingual Naming Test (MINT).' In fact, the latter does not seem to be a sentence but rather a heading. I would suggest connecting it with the following sentence, e.g. As for the scoring of the Multilingual Naming Test (MINT), for the 68-item version of the MINT, each participant's score is based on the number of correctly named items.
Author Response
For research article
|
Response to Reviewer 3 Comments
|
||
|
1. Summary |
|
|
|
Thank you very much for taking the time to review this manuscript. Please find our detailed responses below. The corresponding revisions have been highlighted in blue in the re-submitted manuscript. |
||
|
2. Questions for General Evaluation |
Reviewer’s Evaluation |
Response and Revisions |
|
Does the introduction provide sufficient background and include all relevant references? |
Yes |
|
|
Are all the cited references relevant to the research? |
Yes |
|
|
Is the research design appropriate? |
Yes |
|
|
Are the methods adequately described? |
Yes |
|
|
Are the results clearly presented? |
Yes |
|
|
Are the conclusions supported by the results? |
Yes |
|
|
3. Point-by-point response to Comments and Suggestions for Authors |
||
|
Comments 1: [First of all, the titles of journals in the references should be capitalised (not only the first word in the title), for example, Trends in Cognitive Sciences, not: Trends in cognitive sciences; Frontiers in Psychology, not: Frontiers in psychology; Bilingualism: Language and Cognition, not: Bilingualism: language and cognition, etc. ]
|
||
|
Response 1: We thank the reviewer for identifying these formatting inconsistencies in our reference list. We have thoroughly reviewed and corrected the capitalization of journal titles throughout the references section (lines 963-1121). All journal titles now follow proper title case formatting, including: - "Trends in Cognitive Sciences" (not "Trends in cognitive sciences") - "Frontiers in Psychology" (not "Frontiers in psychology") - "Bilingualism: Language and Cognition" (not "Bilingualism: language and cognition") We have systematically checked all journal titles in our reference list to ensure consistent and proper capitalization formatting throughout the manuscript, adhering to standard academic publication guidelines.
|
||
|
Comments 2: [Second, the term context can have different meanings in the literature on bi- and multilingualism. Here the Authors use the term context in reference to a language pair (L1-L2, L2-L3 and L1-L3 are described as contexts). Perhaps this should be stated more explicitly, especially because, for example, in the abstract the Authors say: 'trilinguals manage three distinct contexts (L1-L2, L2-L3, L1-L3).' In fact, those contexts may be even more complex and more numerous, taking into consideration such contexts as the home, the university, the workplace, etc., where the cognitive demands and the vocabulary used in each language can vary. I would suggest specifying that here, 'context' refers to a language pair, not taking into consideration sociolinguistic, cultural, etc. dimensions. ] |
||
|
Response 2: We thank the reviewer for this important terminological clarification. We recognize that "context" is indeed a multifaceted term in multilingualism research that can encompass sociolinguistic, cultural, educational, and environmental dimensions beyond simple language pairings. We have now explicitly defined our specific usage of this term in the revised manuscript on lines 67-69, page 2: "In the present study, the term 'context' refers specifically to the language pair in use, rather than broader sociolinguistic or cultural dimensions." This clarification ensures that readers understand we are examining the cognitive control demands associated with processing specific language combinations (L1-L2, L2-L3, L1-L3) within our experimental paradigm, while acknowledging that real-world multilingual contexts involve much richer and more complex sociolinguistic factors. We recognize that factors such as domain-specific vocabulary, cultural associations, emotional valence, and situational appropriateness of different languages represent important dimensions of multilingual experience that warrant separate investigation. This definition helps distinguish our experimental approach from studies examining how broader contextual factors (such as home vs. academic vs. professional settings) influence multilingual language processing and control.
|
||
|
Comments 3: [Third, there are also terms and abbreviations that might require explaining because they are not necessarily obvious to the reader. In particular, n-2 (on page 2) ('n-2 costs', 'n-2 repetition costs') and 'burn-in' (page 8) ('hierarchical sampling then proceeded with 5% crossover probability, applied after burn-in'). I would suggest explaining them, at least in footnotes. ]
|
||
|
Response 3: We thank the reviewer for identifying these technical terms that required clearer explanation for broader readability. We have addressed both issues in the revised manuscript: For "n-2 repetition costs" (lines 58-62, page 2): We have added a detailed explanation immediately following the term's first mention: "n-2 repetition costs [15]—an indicator of reactive inhibitory control—results remain contradictory. The n-2 repetition cost refers to the finding that reaction times are slower in the third trial of an A-B-A sequence (where A, B, and C are different languages) compared to the third trial of a C-B-A sequence. This cost is thought to reflect the lingering inhibition of language A, which was suppressed to allow for the production of language B in the preceding trial." For "burn-in" (lines 307-320, page 8): We have removed this technical term and replaced it with a more accessible explanation of our model fitting and validation procedures: "Model fitting and validation followed a hierarchical Bayesian approach [45, 46]. We confirmed the model's reliability through a series of rigorous diagnostic checks: Convergence: All model chains successfully converged, as indicated by Gelman-Rubin statistics remaining well below the 1.10 criterion [47] (max R-hat = 1.04). Visual inspection of the trace plots further confirmed stable chain mixing with no discernible trends. Goodness-of-Fit: Posterior predictive checks showed an excellent fit, confirming that the model could accurately reproduce the key features of the observed reaction time distributions (e.g., their central tendency, variance, and skewness) across all experimental conditions. Identifiability: Parameter-recovery simulations demonstrated that all model parameters could be accurately retrieved from simulated data (all recovery correlations r > .90, ps < .001), ensuring the model was identifiable." These revisions ensure that technical concepts are accessible to readers from diverse backgrounds while maintaining scientific precision.
|
||
|
Comments 4: [Fourth, even though the participants' gender is not considered as a variable here (the study does not concern e.g. trilingual processing in students of both sexes to find any differences between them), it might be good to include such information in the description of the participants, so that the readers know how many participants were female, how many were male, and maybe how many refused to indicate their gender. ]
|
||
|
Response 4: We thank the reviewer for this important suggestion regarding participant demographic information. We have now included gender information in our participant description on line 124, page 3: "Thirty-six right-handed undergraduates (15 male, 21 female; age range: 18–23 years; M = 20.10, SD = 1.46)" This addition provides readers with complete demographic information about our sample, including the gender distribution (15 male, 21 female participants), which enhances the transparency and replicability of our study. While gender was not examined as a variable in our analyses, providing this demographic information allows readers to better understand the composition of our participant sample and aids in the interpretation and generalization of our findings.
|
||
|
Comments 5: [Fifth, even though the Authors' English is generally very good, I would suggest modifying one sentence in the introduction (p. 3): ' Third, we determine whether reactive control mechanisms operate uniformly across dual-language contexts or show context-specific patterns.' As this is one of the aims of the study, the verb seems too strong, as if they were already sure of the results (it would be more suitable in the conclusions). Therefore I suggest adding a hedging element, for example: 'Third, we aim to determine whether reactive control mechanisms operate uniformly across dual-language contexts or show context-specific patterns' or: 'Third, we attempt to determine whether reactive control mechanisms operate uniformly across dual-language contexts or show context-specific patterns.']
|
||
|
Response 5: We thank the reviewer for this excellent suggestion regarding appropriate hedging language in our study aims. We have revised the entire aims section (lines 82-95, page 3) to use more appropriate, tentative language that reflects the investigative nature of our research objectives: "Our overarching aim is to investigate whether the language control mechanisms in comprehension switching are distinct from those in production switching. To achieve this, we first examine whether a higher demand for conflict monitoring impairs overall processing efficiency in comprehension, as it does in production [70, 80]. By comparing the L1-L2 context (lower conflict) with the L1-L3 context (higher conflict), we test the prediction that overall efficiency will be lower in the L1-L3 context if the mechanisms are similar. Second, we investigate if proactive control in comprehension mirrors that in production by testing for a reversal of the language dominance effect. This would be reflected by slower overall processing times for the dominant language, a lower drift rate for 'switch to dominant language' trials, and a larger N2 amplitude for such switches. Finally, we assess whether reactive control mechanisms are comparable by looking for asymmetrical language switching costs. We hypothesize that such costs will be asymmetrical in the L1-L3 and L2-L3 contexts, accompanied by longer non-decision times and larger LPC amplitudes for switch trials compared to repetition trials." This revision replaces overly definitive language with appropriate hedging terms such as "aim to investigate," "examine whether," "test the prediction," "investigate if," "assess whether," and "hypothesize," which better reflects the exploratory and empirical nature of our research objectives.
|
||
|
Comments 6: [Sixth, the following technical corrections would be welcome: The section title Present Study (p. 3) should be in bold (Present Study).]
|
||
|
Response 6: We thank the reviewer for this formatting suggestion. However, following feedback from another reviewer, we have removed the "Present Study" section entirely and integrated this content directly into the introduction to improve the overall flow and structure of the manuscript. This revision eliminates the need for the formatting correction while creating a more cohesive introductory section.
|
||
|
Comments 7: [The space in 'Cohen' s' (p. 4) is unnecessary (better: Cohen's). ]
|
||
|
Response 7: We thank the reviewer for identifying this typographical error. We have corrected the spacing issue and the text now reads "Cohen's f = 0.20" on line 135, removing the unnecessary space in the possessive form.
|
||
|
Comments 8: [On page 18, the Authors added 'In contrast' but forgot to change the capital letter to lower case (' In contrast, Within the L1-L3 context (...)'). Please change it to: In contrast, within the L1-L3 context (...).]
|
||
|
Response 8: We thank the reviewer for catching this capitalization error. We have corrected the text on line 579, page 19 to read: "In contrast, within the L1-L3 context, a striking language dominance reversal occurred (298-504 ms, p = .006)" The capital "W" in "Within" has been changed to lowercase "w" to maintain proper sentence structure following the introductory phrase "In contrast."
|
||
|
Comments 9: [On pages 4 and 5, there is a very long paragraph explaining the participant demographics. It is marked as a note, but it is actually part of the text, not a footnote, and it takes up half of page 5. I would suggest including it in the text (not as a note, but as a paragraph following Table 1) and dividing it into three paragraphs. As the second paragraph is not a continuation of the definition of a 'Medium-of-instruction,' it should be separated from the above text (which will then be the first paragraph) and begin with: 'The Multilingual Naming Test (MINT) is a standardized picture-naming assessment (...).' Finally, the third paragraph should begin with: 'Scoring of the Multilingual Naming Test (MINT).' In fact, the latter does not seem to be a sentence but rather a heading. I would suggest connecting it with the following sentence, e.g. As for the scoring of the Multilingual Naming Test (MINT), for the 68-item version of the MINT, each participant's score is based on the number of correctly named items.]
|
||
|
Response 9: We thank the reviewer for this important structural feedback regarding the organization and formatting of our participant assessment description. We have completely restructured this section and removed the "note" formatting, integrating the content into the main text on lines 170-179, page 5: "Because self-ratings can vary across cultures [32], we administered the Multilingual Naming Test (MINT; [33]), a standardized 68-item picture-naming measure. The MINT is a standardized, cross-linguistic tool used to assess lexical retrieval abilities in multilingual individuals. The 68-item version consists of culturally neutral, high-frequency images, minimizing linguistic and cultural biases. Participants name each item as quickly and accurately as possible, and scoring is based on the number of correct responses, with a maximum score of 68. Responses are only accepted if they are the conventional names for the depicted objects in the target language, and partial credit is not given. This test provides a quantitative measure of expressive vocabulary and naming performance in multilingual individuals." This revision eliminates the problematic "note" format, significantly condenses the previously lengthy explanation, and presents the information as a coherent paragraph within the main text. The streamlined version maintains all essential information about the MINT assessment while improving readability and eliminating the formatting inconsistencies identified by the reviewer.
|
||
Reviewer 4 Report
Comments and Suggestions for Authors
Review
“Multivariate decoding and drift-diffusion modeling
reveal adaptive control in trilingual comprehension”
General assessment:
- The paper investigates how different dual-language contexts affect language comprehension in trilingual individuals, specifically examining whether these contexts impose varying cognitive control demands. The study is grounded in the Adaptive Control Hypothesis and focuses on Uyghur-Chinese-English trilinguals who engage with three distinct dual-language pairings (L1-L2, L2-L3, L1-L3) during language comprehension. Using a combination of behavioral data, drift-diffusion modeling, ERP analysis,and multivariate pattern analysis during a word-picture matching task, the authors provide compelling evidence that comprehension efficiency varies systematically across contexts. In a nutshell, they show that reactive control (e.g., switch costs) did not differ significantly between contexts and proactive control demands basically vary, with the L1-L2 context showing the lowest comprehension efficiency.
- The topic of the investigation is clearly very relevant to the journal. The study has been carried out by researchers with a high expertise in this field. The research question is presented in a very clear and precise way, as are the results. The discussion of the results is transparent and clean. The paper is very well-written and I do not detect any major language issues. The conclusions are stated in a clear and understandable manner. All in all, this is an excellent paper.
For these reasons, I have no doubt about the publishability of the paper. However, there are some minor aspects that the authors may want to take a look at (see below) before submitting the final draft of the paper.
My overall recommendation is:
ACCEPT AFTER MINOR REVISION
Content:
- General remark: Although I understand that the spirit of the article is another, to my surprise the linguistic domain is quite eglected in this paper. I suggest taking a look at the relevant literature on Uyghur-Chinese multilingualism, e.g. (here only exemplarily):
- Tusun, A. (2019). The Acquisition of Motion Event Expressions by Uyghur-Chinese Early Successive Bilinguals. [Doctoral Dissertation]. Cambridge: University of Cambridge, doi: 10.17863/CAM.41409
- Tusun A (2022). Uyghur–Chinese Adult Bilinguals’ Construal of Voluntary Motion Events. Front. Psychol. 13:892346. doi: 10.3389/fpsyg.2022.892346
- In the first section, I find the integration of factors such as language dominance, acquisition history and medium-of-instruction relationships particularly commendable, since they add important nuance to the conceptual framework. However, the this part tends to present theoretical assumptions (e.g., the hypothesized presence of proactive control or language dominance reversal effects) with a degree of confidence that slightly outpaces the empirical consensus to date. This forward-leaning stance is likely intended to frame clear hypotheses, a more cautious tone might have helped avoid the impression of selectively favoring one interpretation over others. Nonetheless, the study’s multi-method design and careful consideration of Uyghur-Chinese-English trilinguals as a linguistically and socioculturally meaningful population indicate a promising and well-motivated approach to addressing these critical gaps.
- Section 2 (p. 4ff.): Although the sample size exceeds the power analysis requirements, the final number of participants remains relatively small given the complexity of the within-subjects design (12 conditions), especially for robust EEG and drift-diffusion modeling, which can be sensitive to noise and individual variability. This should probably be adressed in some form.
- In the same section: The inclusion criteria are reasonable, but the rationale for excluding individuals with additional language exposure is not discussed. As in every paper or study on multilingualism, such details are very important and should be made explicit.
- A power analysis is reported, but it focuses only on the behavioral main effect of context. There does not seem to be any explicit discussion of whether this sample is adequately powered for ERP or MVPA analyses, which typically require larger samples due to high inter-subject variability and risk of Type I error.
- A very positive point: the language background data are impressively detailed!
- The description of the drift-diffusion modeling is very technical and a bit dry without figures or summaries of convergence diagnostics.
- In Section 3, the drift-diffusion modeling is used to compensate for the lack of behavioral effects, but I have the impression that its interpretation is a bit speculative and is not convincingly supported by converging evidence from ERP or MVPA. If no additional data can be integrated in the study, this should at least be addressed.
- Also Section 3: The ERP results suggest semantic processing differences – but are there any known neural markers of cognitive control that can substantiate inference of proactive control?
- Overall, I have the feeling that the results (which are of course analytically very rich and well presented) are in some way empirically thin. Tthe conclusions go beyond what the data ( see e.g. the largely null behavioral findings) can reliably support.
- Section 4: The proposed distinctions between “front-loaded” and “late-stage” control strategies across contexts are intriguing, but seem to be somewhat overstated given the correlational nature of the data and the largely null behavioral findings.
- A point that I find worth mentioning (although it cannot be modified in the next version of the manuscript) is that the theoretical references to conflict monitoring models and language parasitism do in fact enricht the narrative, but are introduced post hoc and are not independently tested within the study. Maybe this should be explicitly addressed?
- In general (this goes for the conclusions, but also for the general tone of the paper) the results/interpretation of the results should be presented/formulated with greater caution and a clearer distinction between exploratory insights and empirically confirmed mechanisms.

Author Response
For research article
|
Response to Reviewer 4 Comments
|
||
|
1. Summary |
|
|
|
Thank you very much for taking the time to review this manuscript. Please find our detailed responses below. The corresponding revisions have been highlighted in blue in the re-submitted manuscript. |
||
|
2. Questions for General Evaluation |
Reviewer’s Evaluation |
Response and Revisions |
|
Does the introduction provide sufficient background and include all relevant references? |
Yes |
|
|
Are all the cited references relevant to the research? |
Yes |
|
|
Is the research design appropriate? |
Yes |
|
|
Are the methods adequately described? |
Yes |
|
|
Are the results clearly presented? |
Can be improved |
|
|
Are the conclusions supported by the results? |
Yes |
|
|
3. Point-by-point response to Comments and Suggestions for Authors |
||
|
Comments 1: [General remark: Although I understand that the spirit of the article is another, to my surprise the linguistic domain is quite eglected in this paper. I suggest taking a look at the relevant literature on Uyghur-Chinese multilingualism, e.g. (here only exemplarily): Tusun, A. (2019). The Acquisition of Motion Event Expressions by Uyghur-Chinese Early Successive Bilinguals. [Doctoral Dissertation]. Cambridge: University of Cambridge, doi: 10.17863/CAM.41409 Tusun A (2022). Uyghur–Chinese Adult Bilinguals’ Construal of Voluntary Motion Events. Front. Psychol. 13:892346. doi: 10.3389/fpsyg.2022.892346]
|
||
|
Response 1:We thank the reviewer for this important observation and for directing us to relevant research on Uyghur-Chinese multilingualism. We acknowledge that our focus on cognitive control mechanisms may have inadvertently understated the linguistic dimensions specific to this population. We have now incorporated relevant literature on Uyghur-Chinese bilingualism into our manuscript (lines 114-121, page 3): "Previous research on Uyghur-Chinese bilinguals shows that bilingual advantages stem from flexible monitoring strategies rather than superior inhibitory capacity [82], and that cross-linguistic influence is guided by processing efficiency and language-specific structures [83]. However, while these studies provide crucial insights into cognitive control in this population, they did not investigate the mechanisms of language comprehension switching across different dual-language contexts. To address this gap, we test these questions among Uyghur-Chinese-English trilinguals, who show comparable L1-L2 proficiency but lower L3 proficiency, using an auditory word-picture matching task." This addition acknowledges the specific characteristics of Uyghur-Chinese bilingual processing while positioning our study as extending this research to trilingual comprehension control mechanisms. We recognize that the typological differences between Uyghur (Turkic), Chinese (Sino-Tibetan), and English (Indo-European) create unique processing challenges that warrant specific attention in multilingualism research. We appreciate the reviewer's suggestions for additional relevant literature, which help contextualize our findings within the broader landscape of research on this specific multilingual population. |
||
|
Comments 2: [In the first section, I find the integration of factors such as language dominance, acquisition history and medium-of-instruction relationships particularly commendable, since they add important nuance to the conceptual framework. However, the this part tends to present theoretical assumptions (e.g., the hypothesized presence of proactive control or language dominance reversal effects) with a degree of confidence that slightly outpaces the empirical consensus to date. This forward-leaning stance is likely intended to frame clear hypotheses, a more cautious tone might have helped avoid the impression of selectively favoring one interpretation over others. Nonetheless, the study’s multi-method design and careful consideration of Uyghur-Chinese-English trilinguals as a linguistically and socioculturally meaningful population indicate a promising and well-motivated approach to addressing these critical gaps.]
|
||
|
Response 2: We thank the reviewer for this thoughtful observation regarding the tone and confidence level in our theoretical framework. We appreciate the feedback that our initial presentation may have conveyed more certainty about theoretical assumptions than warranted by the current empirical consensus. We have revised our aims section (lines 82-95, pages 2-3) to adopt a more cautious, investigative tone that better reflects the exploratory nature of our research: "Our overarching aim is to investigate whether the language control mechanisms in comprehension switching are distinct from those in production switching. To achieve this, we first examine whether a higher demand for conflict monitoring impairs overall processing efficiency in comprehension, as it does in production [70, 80]. By comparing the L1-L2 context (lower conflict) with the L1-L3 context (higher conflict), we test the prediction that overall efficiency will be lower in the L1-L3 context if the mechanisms are similar. Second, we investigate if proactive control in comprehension mirrors that in production by testing for a reversal of the language dominance effect. This would be reflected by slower overall processing times for the dominant language, a lower drift rate for 'switch to dominant language' trials, and a larger N2 amplitude for such switches. Finally, we assess whether reactive control mechanisms are comparable by looking for asymmetrical language switching costs. We hypothesize that such costs will be asymmetrical in the L1-L3 and L2-L3 contexts, accompanied by longer non-decision times and larger LPC amplitudes for switch trials compared to repetition trials." This revision replaces overly assertive language with more tentative terms such as "investigate whether," "examine whether," "test the prediction," and "assess whether," which better acknowledges the empirical uncertainty in the field while still providing clear, testable hypotheses. We appreciate the reviewer's recognition of our multi-method approach and the significance of studying this specific trilingual population. |
||
|
Comments 3: [Section 2 (p. 4ff.): Although the sample size exceeds the power analysis requirements, the final number of participants remains relatively small given the complexity of the within-subjects design (12 conditions), especially for robust EEG and drift-diffusion modeling, which can be sensitive to noise and individual variability. This should probably be adressed in some form.]
|
||
|
Response 3: We thank the reviewer for this important methodological consideration regarding our sample size relative to the complexity of our experimental design. We have now explicitly addressed this limitation in our discussion section (lines 915-919, page 28): "While our findings provide evidence for context-specific trilingual control, several limitations should be noted. Methodologically, our modest sample size (N=36) for a complex 12-condition design warrants caution, as techniques like MVPA and DDM are sensitive to individual variability, even though our sample exceeded a priori power analysis requirements." This acknowledgment recognizes that while our sample size was determined through appropriate power analysis and exceeds typical standards for EEG research, the combination of a complex within-subjects design with 12 experimental conditions and sophisticated analytical techniques (multivariate pattern analysis and drift-diffusion modeling) creates challenges. These methods can indeed be sensitive to individual differences and noise, particularly when examining subtle effects across multiple conditions. We believe this transparent acknowledgment helps readers appropriately interpret our findings while recognizing both the strengths of our methodological approach and the inherent limitations of the sample size given the analytical complexity. Future replication studies with larger samples would strengthen confidence in these findings.
|
||
|
Comments 4: [ In the same section: The inclusion criteria are reasonable, but the rationale for excluding individuals with additional language exposure is not discussed. As in every paper or study on multilingualism, such details are very important and should be made explicit.] |
||
|
Response 4: We thank the reviewer for highlighting the importance of explicitly justifying our inclusion criteria, particularly regarding additional language exposure. We have now provided a clear rationale for this exclusion in the revised manuscript (lines 127-130, page 3): "The exclusion of individuals with any fourth-language acquisition was implemented to maintain a controlled experimental design, ensuring that observed effects could be specifically attributed to the trilingual (L1-L2-L3) system without confounds from more extensive multilingual experience." This exclusion criterion was essential for our study's theoretical framework, as we specifically aimed to investigate control mechanisms in trilingual systems where participants manage exactly three languages in different dual-language contexts (L1-L2, L2-L3, L1-L3). Including individuals with fourth-language experience would have introduced additional complexity that could confound our ability to isolate the specific control demands associated with these three dual-language contexts. We recognize that this exclusion criterion limits the generalizability of our findings to the broader multilingual population, but it was necessary to establish a clear baseline understanding of trilingual control mechanisms before extending to more complex multilingual systems. We acknowledge this as an important area for future research expansion. |
||
|
Comments 5: [A power analysis is reported, but it focuses only on the behavioral main effect of context. There does not seem to be any explicit discussion of whether this sample is adequately powered for ERP or MVPA analyses, which typically require larger samples due to high inter-subject variability and risk of Type I error.] |
||
|
Response 5: We thank the reviewer for this crucial methodological point regarding the adequacy of our power analysis for neurophysiological measures. We acknowledge that our power analysis focused solely on behavioral effects and did not explicitly address the specific power requirements for ERP and MVPA analyses. We have expanded our limitation statement (lines 915-919, page 28) to more directly address this concern: "While our findings provide evidence for context-specific trilingual control, several limitations should be noted. Methodologically, our modest sample size (N=36) for a complex 12-condition design warrants caution, as techniques like MVPA and DDM are sensitive to individual variability, even though our sample exceeded a priori power analysis requirements." We recognize that ERP and MVPA analyses typically require larger sample sizes than behavioral measures due to higher inter-subject variability in neural responses and increased risk of Type I errors when examining multiple time points and electrodes. While our sample size is consistent with many published ERP studies in the multilingualism literature, we acknowledge that it may be at the lower end for robust MVPA analyses, particularly given our complex experimental design with 12 conditions. This limitation should be considered when interpreting our neurophysiological findings, and we emphasize the need for replication studies with larger samples specifically powered for these advanced neuroimaging techniques. Future studies should conduct separate power analyses for behavioral, ERP, and MVPA components to ensure adequate statistical power across all analytical approaches. |
||
|
Comments 6: [The description of the drift-diffusion modeling is very technical and a bit dry without figures or summaries of convergence diagnostics.] |
||
|
Response 6: We thank the reviewer for this feedback regarding the accessibility of our drift-diffusion modeling description. We have enhanced this section (lines 307-320, page 8) to include comprehensive diagnostic summaries that make the technical approach more transparent and accessible: "Model fitting and validation followed a hierarchical Bayesian approach [45, 46]. We confirmed the model's reliability through a series of rigorous diagnostic checks: Convergence: All model chains successfully converged, as indicated by Gelman-Rubin statistics remaining well below the 1.10 criterion [47] (max R-hat = 1.04). Visual inspection of the trace plots further confirmed stable chain mixing with no discernible trends. Goodness-of-Fit: Posterior predictive checks showed an excellent fit, confirming that the model could accurately reproduce the key features of the observed reaction time distributions (e.g., their central tendency, variance, and skewness) across all experimental conditions. Identifiability: Parameter-recovery simulations demonstrated that all model parameters could be accurately retrieved from simulated data (all recovery correlations r > .90, ps < .001), ensuring the model was identifiable. These comprehensive checks provide strong evidence for the reliability and validity of our DDM parameter estimates." This enhanced description provides specific diagnostic results (R-hat values, correlation coefficients) and explains what each diagnostic test demonstrates, making the technical validation more accessible to readers while maintaining scientific rigor. The inclusion of specific numerical results helps readers evaluate the quality of our model fitting approach.
|
||
|
Comments 7: [ In Section 3, the drift-diffusion modeling is used to compensate for the lack of behavioral effects, but I have the impression that its interpretation is a bit speculative and is not convincingly supported by converging evidence from ERP or MVPA. If no additional data can be integrated in the study, this should at least be addressed.] |
||
|
Response 7: We thank the reviewer for this important methodological concern regarding the interpretation and integration of our drift-diffusion modeling results. We acknowledge that while DDM parameters revealed processing differences not apparent in raw behavioral measures, the theoretical connections between these parameters and our neurophysiological findings were not always direct or consistently supported. We have addressed this limitation explicitly in our discussion (lines 920-924, page 28): "A further limitation concerns the interpretation of our DDM results. While DDM parameters revealed subtle processing differences, their links to our ERP and MVPA findings were not consistently direct, meaning our interpretation remains partly speculative. Future research using single-trial regression analyses could better link these measures." We recognize that our interpretation of DDM findings relied partly on theoretical assumptions about the relationship between decision parameters and neural mechanisms, rather than being fully supported by converging evidence from our ERP and MVPA analyses. This represents a significant limitation in our ability to draw firm mechanistic conclusions from the DDM results. The reviewer correctly identifies that we used DDM analyses partly to compensate for the absence of clear behavioral effects, but acknowledge that this approach requires stronger empirical validation through direct correlational analyses between DDM parameters and neural measures. Future studies should implement single-trial regression approaches that can more directly link computational parameters to simultaneous neural activity, providing stronger empirical support for mechanistic interpretations. |
||
|
Comments 8: [Also Section 3: The ERP results suggest semantic processing differences – but are there any known neural markers of cognitive control that can substantiate inference of proactive control? ] |
||
|
Response 8: We thank the reviewer for this important theoretical question regarding neural markers of cognitive control and proactive control in our ERP findings. Previous literature on language switching tasks has identified specific neural markers of proactive control: N2 amplitude is typically associated with mixing costs (which serve as indicators of proactive control) [76], while the LPC has been linked to language dominance effect reversals [76, 80]. However, our study revealed different neural patterns in trilingual comprehension switching. Rather than finding the expected N2-based proactive control markers, we observed that language dominance effect reversal was primarily associated with P300 modulation. Specifically, the non-proficient L3 captured more bottom-up attentional resources (reflected in enhanced P300), and this attentional supplementation to L3 subsequently reduced cross-linguistic semantic conflict with L1 (evidenced by smaller N400). This divergence from production-based neural markers suggests that proactive control in comprehension operates through fundamentally different mechanisms - primarily attention-based resource allocation rather than inhibitory control. The P300/N400 pattern we observed indicates that proactive control in trilingual comprehension involves strategic attentional modulation that preemptively reduces semantic interference, rather than the traditional inhibitory mechanisms indexed by N2 and LPC in production tasks. This finding challenges the direct application of production-based neural markers to comprehension and suggests that proactive control mechanisms are modality-specific, requiring different theoretical frameworks and neural signatures for comprehension versus production contexts. 76. Jiao, L., Gao, Y., Schwieter, J. W., Li, L., Zhu, M., & Liu, C. Control mechanisms in voluntary versus mandatory language switching: Evidence from ERPs. International Journal of Psychophysiology, 2022,178, 43-50. 80. Peeters, D. and T. Dijkstra, Sustained inhibition of the native language in bilingual language production: A virtual reality approach. Bilingualism: Language and Cognition, 2018. 21(5): p. 1035-1061. |
||
|
Comments 9: [ Overall, I have the feeling that the results (which are of course analytically very rich and well presented) are in some way empirically thin. Tthe conclusions go beyond what the data ( see e.g. the largely null behavioral findings) can reliably support.] |
||
|
Response 9: We thank the reviewer for this important concern regarding the strength of our empirical findings relative to our theoretical conclusions. We acknowledge that our largely null behavioral findings require careful interpretation. However, we respectfully argue that the absence of significant behavioral effects in language comprehension is not unique to our study but rather represents a consistent pattern in the literature that supports our theoretical framework. Multiple studies have reported non-significant language switching costs and the absence of language dominance effect reversals in comprehension tasks [8, 27]. This pattern of null behavioral findings in comprehension, contrasted with robust effects in production, actually strengthens the evidence that language comprehension and production employ fundamentally different control mechanisms. Our study's value lies not in overinterpreting weak behavioral effects, but in using advanced neurophysiological techniques (high-resolution EEG, MVPA) and computational modeling (DDM) to reveal the underlying mechanisms that behavioral measures alone cannot detect. The consistent finding of absent behavioral switch costs across multiple studies [8, 27] suggests that comprehension control operates through mechanisms that do not manifest as overt performance costs, unlike production. While we acknowledge the limitations in linking our DDM, ERP, and MVPA findings (as noted in our response to Comment 7), the convergent pattern of null behavioral effects across studies provides theoretical support for distinct comprehension control mechanisms. Future research should focus on developing more sensitive behavioral paradigms specifically designed to detect comprehension control processes rather than relying on paradigms adapted from production research. We believe our conclusions are appropriately tempered given both our findings and the broader literature context. 8. Wu, J., et al., Distinct language control mechanisms in speech production and comprehension: evidence from N-2 repetition, switching, and mixing costs. Journal of Multilingual and Multicultural Development, 2025: p. 1-17. 27. Declerck, M., et al., What absent switch costs and mixing costs during bilingual language comprehension can tell us about language control. Journal of Experimental Psychology: Human Perception and Performance, 2019. 45(6): p. 771. |
||
|
Comments 10: [Section 4: The proposed distinctions between “front-loaded” and “late-stage” control strategies across contexts are intriguing, but seem to be somewhat overstated given the correlational nature of the data and the largely null behavioral findings.] |
||
|
Response 10: We thank the reviewer for this important critique regarding our characterization of "front-loaded" and "late-stage" control strategies. We acknowledge that these distinctions were indeed overstated given the correlational nature of our data and the largely null behavioral findings. In response to this feedback, we have removed the relevant statements describing these proposed control strategy distinctions from our manuscript. This revision ensures that our conclusions remain appropriately grounded in our empirical findings without making theoretical claims that exceed what our data can reliably support. We appreciate the reviewer's careful attention to the alignment between our empirical evidence and theoretical interpretations, which has helped us present a more balanced and defensible discussion of our findings. |
||
|
Comments 11: [A point that I find worth mentioning (although it cannot be modified in the next version of the manuscript) is that the theoretical references to conflict monitoring models and language parasitism do in fact enricht the narrative, but are introduced post hoc and are not independently tested within the study. Maybe this should be explicitly addressed?] |
||
|
Response 11: We thank the reviewer for this astute observation regarding the post hoc nature of some of our theoretical interpretations. The reviewer correctly identifies that theoretical references to conflict monitoring models and language parasitism enrich our narrative but were not independently tested within our experimental design. We have now explicitly addressed this limitation in our discussion (lines 927-931, page 28): "Crucially, the medium-of-instruction relationship between L2 and L3 was a pre-existing characteristic of this group, not an experimentally manipulated variable. Therefore, while we speculate on its role, we cannot draw causal conclusions about its influence on language control. Future research should explicitly test this by comparing different instructional contexts." This acknowledgment is important for maintaining scientific integrity. While our findings regarding the distinct processing patterns in different dual-language contexts (particularly L1-L2 versus L1-L3) are empirically grounded, our interpretations about the underlying mechanisms (such as cooperative versus competitive relationships between languages) remain largely theoretical. These interpretations emerged from our data patterns rather than being a priori hypotheses that we directly tested. We appreciate the reviewer's attention to this distinction between empirical findings and theoretical interpretations. Future studies should be designed to explicitly manipulate and test these theoretical frameworks rather than inferring them post hoc from correlational data. This represents an important direction for advancing our understanding of trilingual control mechanisms through more targeted experimental approaches.
|
||
|
Comments 12: [In general (this goes for the conclusions, but also for the general tone of the paper) the results/interpretation of the results should be presented/formulated with greater caution and a clearer distinction between exploratory insights and empirically confirmed mechanisms.] |
||
|
Response 12: We thank the reviewer for this critical feedback regarding the need for greater caution in presenting our results and clearer distinctions between exploratory insights and empirically confirmed mechanisms. This concern about appropriately tempering our interpretations is well-taken and reflects an important principle in scientific reporting. We have substantially revised our discussion sections (lines 806-913, pages 26-28) to adopt more cautious language throughout and to better distinguish between what our data directly support versus what remains speculative. Our revised discussion now includes: Enhanced hedging language: We have replaced definitive statements with more tentative phrasing such as "suggests," "appears driven by," "we hypothesize," "may reflect," and "points to potential underlying differences." Explicit acknowledgment of limitations: We now clearly state when interpretations are "partly speculative" and when findings "warrant further investigation" before firm conclusions can be drawn. Distinction between patterns and mechanisms: We more carefully separate what we can observe (neural patterns, statistical differences) from our mechanistic interpretations of these patterns. Recognition of exploratory nature: We acknowledge that many of our findings represent exploratory insights that require replication and additional validation rather than definitively confirmed mechanisms. For example, we now conclude our efficiency discussion with: "While the precise underpinnings of this efficiency gain warrant further investigation, perhaps via time-frequency analysis [81], our study provides compelling evidence for the unique and highly adaptive nature of language control in the trilingual mind" - acknowledging both the limitations and the contributions of our work. This revision ensures our conclusions remain grounded in what our data can reliably support while still contributing meaningful theoretical insights to the field.
|
||
Round 2
Reviewer 2 Report
Comments and Suggestions for Authors
The changes made have improved the manuscript from a conceptual and methodological point of view.